# Optimization for Approximate Submodularity

**Avinatan Hassidim**
Bar Ilan University and Google
`avinatan@cs.biu.ac.il`

**Yaron Singer**
Harvard University
`yaron@seas.harvard.edu`

## Abstract

We consider the problem of maximizing a submodular function when given access to its approximate version. Submodular functions are heavily studied in a wide variety of disciplines since they are used to model many real world phenomena and are amenable to optimization. There are many cases however in which the phenomena we observe is only *approximately* submodular and the optimization guarantees cease to hold. In this paper we describe a technique that yields strong guarantees for maximization of monotone submodular functions from approximate surrogates under cardinality and intersection of matroid constraints. In particular, we show tight guarantees for maximization under a cardinality constraint and $1/(1 + P)$ approximation under intersection of $P$ matroids.

## 1   Introduction

In this paper we study maximization of *approximately* submodular functions. For nearly half a century submodular functions have been extensively studied since they are amenable to optimization and are recognized as an effective modeling instrument. In machine learning, submodular functions capture a variety of objectives that include entropy, diversity, image segmentation, and clustering.

Although submodular functions are used to model real world phenomena, in many cases the functions we encounter are only *approximately* submodular. This is either due to the fact that the objectives are not exactly submodular, or alternatively they are, but we only observe their approximate version.

In the literature, approximate utility functions are modeled as surrogates of the original function that have been corrupted by random variables drawn i.i.d from some distribution. Some examples include:

- **Revealed preference theory.** Luce's famous model assumes that an agent's revealed utility $\tilde{f} : 2^N \to \mathbb{R}$ can be approximated by a well-behaved utility function $f : 2^N \to \mathbb{R}$ s.t. $\tilde{f}(S) = f(S) + \xi_S$ for every $S \subseteq N$ where $\xi_S$ is drawn i.i.d from a distribution [CE16]. $\tilde{f}$ is a utility function that approximates $f$, and multiple queries to $\tilde{f}$ return the same response;

- **Statistics and learning theory.** The assumption in learning is that the data we observe is generated by $\tilde{f}(\mathbf{x}) = f(\mathbf{x}) + \xi_{\mathbf{x}}$ where $f$ is in some well-behaved hypothesis class and $\xi_{\mathbf{x}}$ is drawn i.i.d from some distribution. The use of $\tilde{f}$ is not to model corruption by noise but rather the fact that data is not *exactly* manufactured by a function in the hypothesis class;

- **Active learning.** There is a long line of work on noise-robust learning where one has access to a noisy membership oracle $\tilde{f}(\mathbf{x}) = f(\mathbf{x}) + \xi_{\mathbf{x}}$ and for every $\mathbf{x}$ we have that $\xi_{\mathbf{x}}$ is drawn i.i.d from a distribution [Ang88, GKS90, Jac94, SS95, BF02, Fel09]. In this model as well, the oracle is consistent and multiple queries return the same response. For set functions, one can consider active learning in experimental design applications where the objective function is often submodular and the goal would be to optimize $f : 2^N \to \mathbb{R}$ given $\tilde{f}$.

Similar to the above examples, we say that a function $\tilde{f} : 2^N \to \mathbb{R}_+$ is *approximately submodular* if there is a submodular function $f : 2^N \to \mathbb{R}_+$ and a distribution $\mathcal{D}$ s.t. for each set $S \subseteq N$ we have that $\tilde{f}(S) = \xi_S f(S)$ where $\xi_S$ is drawn i.i.d from $\mathcal{D}$.[1] The modeling assumption that $\xi_S$ is not adversarially chosen but drawn i.i.d is crucial. Without this assumption for $\tilde{f}(S) = \xi_S f(S)$ where $\xi_S \in [1 - \epsilon, 1 + \epsilon]$ even for subconstant $\epsilon > 0$ no algorithm can obtain an approximation strictly better than $n^{-1/2}$ to maximizing either $\tilde{f}$ or $f$ under a cardinality constraint when $n = |N|$ [HS17]. This hardness stems from the fact that approximate submodularity implies that the function is close to submodular, but its marginals (or gradients) are not well approximated by those of the submodular function. When given an $\alpha$ approximation to the marginals, the greedy algorithm produces a $1 - 1/e^\alpha$ approximation. Furthermore, even for *continuous* submodular functions, a recent line of work shows that gradient methods produce strong guarantees with approximate gradient information of the function [HSK17, CHHK18, MHK18].

In contrast to the vast literature on submodular optimization, optimization of approximate submodularity is nascent. For distributions bounded in $[1 - \epsilon/k, 1 + \epsilon/k]$ the function is sufficiently close to submodular for the approximation guarantees of the greedy algorithm to go through [HS, LCSZ17, QSY$^+$17]. Without this assumptions, the greedy algorithm performs arbitrarily poorly. In [HS17] the authors give an algorithm that obtains an approximation arbitrarily close to $1 - 1/e$ under a cardinality constraint that is sufficiently large. For arbitrary cardinality and general matroid constraints there are no known approximation guarantees.

## 1.1 Our Contribution

In this paper we consider the problem $\max_{S \in \mathcal{F}} f(S)$ when $f : 2^N \to \mathbb{R}$ is non-negative monotone submodular defined of a ground set $N$ of size $n$ and the algorithm is only given access to an approximate surrogate $\tilde{f}$ and $\mathcal{F}$ is a uniform matroid (cardinality) or intersection of matroids constraint. We introduce a powerful technique which we call the *sampled mean approximation* and show:

- **Optimal guarantees for maximization under a cardinality constraint.** In [HS17] the result gives an approximation arbitrarily close to $1 - 1/e$ for $k \in \Omega(\log \log n)$. This is a fundamental limitation in their technique that initializes the solution with a set of size $\Omega(\log \log n)$ used for "smoothing" the approximately submodular function (see Appendix G.1 for more details). The technique in this paper is novel and yields an approximation of $1 - 1/e$ for any $k \geq 2$, and $1/2$ for $k = 1$, which is information theoretically tight, as we later show;

- $1/(1 + P)$ **approximation for intersection of $P$ matroids.** We utilize the sampled mean approximation method to produce the first results for the more challenging case of maximization under general matroid constraints. Our approximation guarantees are comparable with those achievable with a greedy algorithm for monotone submodular functions;

- **Information theoretic lower bounds.** We show that no randomized algorithm can obtain an approximation strictly better than $1/2 + \mathcal{O}(n^{-1/2})$ for $\max_{a \in N} f(a)$ given an approximately submodular oracle $\tilde{f}$, and that no randomized algorithm can obtain an approximation strictly better than $(2k - 1)/2k + \mathcal{O}(n^{-1/2})$ for maximization under cardinality constraint $k$;

- **Bounds in Extreme Value Theory.** As we later discuss, some of our results may be of independent interest to Extreme Value Theory (EVT) which studies the bounds on the maximum sample (or the top samples) from some distribution. To achieve our main result we prove subtle properties about extreme values of random variables where not all samples are created equal and the distributions generalize those typically studied in EVT.

The results above are for the problem $\max_{S \in \mathcal{F}} f(S)$ when the algorithm is given access to $\tilde{f}$. In some applications, however, $\tilde{f}$ is the function that we actually wish to optimize, i.e. our goal is to solve $\max_{S \in \mathcal{F}} \tilde{f}(S)$. If $\tilde{f}(S)$ approximates $f(S)$ well on all sets $S$, we can use the solution for $\max_{S \in \mathcal{F}} f(S)$ as a solution for $\max_{S \in \mathcal{F}} \tilde{f}(S)$. In general, however, a solution that is good for $\max_{S \in \mathcal{F}} f(S)$ can be arbitrarily bad for $\max_{S \in \mathcal{F}} \tilde{f}(S)$. In Appendix E we give a black-box reduction showing that these problems are essentially equivalent. Specifically, we show that given a solution

to $\max_{S \in \mathcal{F}} f(S)$ one can produce a solution that is of arbitrarily close quality to $\max_{S \in \mathcal{F}} \tilde{f}(S)$ when $\mathcal{F}$ is any uniform matroid, an intersection of matroids of rank $\Omega(\log n)$, and an intersection of matroids of any rank when the distribution $\mathcal{D}$ has bounded support.

## 1.2 Technical Overview

The approximately submodular functions we consider approximate a submodular function $f$ using samples from a distribution $\mathcal{D}$ of the class of *generalized exponential tail* distributions defined as:

**Definition.** *A noise distribution $\mathcal{D}$ has a **generalized exponential tail** if there exists some $x_0$ such that for $x > x_0$ the probability density function $\rho(x) = e^{-g(x)}$, where $g(x) = \sum_i a_i x^{\alpha_i}$ for some (not necessarily integers) $\alpha_0 \geq \alpha_1 \geq \ldots$, s.t. $\alpha_0 \geq 1$. If $\mathcal{D}$ has bounded support we only require that either it has an atom at its supremum, or that $\rho$ is continuous and non-zero at the supremum.*

This class of distributions is carefully defined. On the one hand it is general enough to contain Gaussian and Exponential distributions, as well as any distribution with bounded support. On the other hand it has enough structure one can leverage. Note that optimization in this setting always requires that the support is independent of $n$ and that $n$ is sufficiently large [2]. Throughout the paper we assume that $\mathcal{D}$ has a generalized exponential tail and that $n$ is sufficiently large.

**Theorem.** *For any non-negative monotone submodular function there is a deterministic polynomial-time algorithm which optimizes the function under a cardinality constraint $k \geq 3$ and obtains an approximation ratio that is arbitrarily close to $1 - 1/e$ with probability $1 - o(1)$ using access to an approximate oracle. For $k \geq 2$ there is a randomized algorithm whose approximation ratio is arbitrarily close to $1 - 1/e$, in expectation over the randomization of the algorithm. For $k = 1$ the algorithm achieves a $1/2$ approximation in expectation, and no randomized algorithm can achieve an approximation better than $1/2 + o(1)$, in expectation.*

The main part of the proof involves analysis of the following greedy algorithm. The algorithm iteratively chooses *bundles* of elements of size $\mathcal{O}(1/\epsilon)$. In each iteration, the algorithm first identifies a bundle $\mathbf{x}$ whose addition to the current solution approximately maximizes the *approximate mean value* $\tilde{F}$. Informally, $\tilde{F}(\mathbf{x})$ is the average value of $\tilde{f}$ evaluated on all bundles at Hamming distance one from $\mathbf{x}$. Then, the algorithm does not choose $\mathbf{x}$ but rather the bundle at Hamming distance one from $\mathbf{x}$ whose addition to the current solution maximizes the approximate submodular value $\tilde{f}$.

The major technical challenge is in analyzing the regime in which $k \in \Omega(1/\epsilon^2) \cap \mathcal{O}(\sqrt{\log n})$. At a high level, in this regime the analysis relies on showing that the marginal contribution of the bundle of elements selected in every iteration is approximately largest. Doing so requires proving subtle properties about extreme values of random variables drawn from the generalized exponential tail distribution, and the analysis fully leverages the properties of the distribution and the fact that $k \in \mathcal{O}(\sqrt{\log n})$. This is of independent interest to Extreme Value Theory (EVT) which tries to bound the maximum sample (or the top samples) from some distribution. If we would consider the constant function $f(S) = 1$ for $S \neq \emptyset$, and try to maximize an approximate version of $f$ with respect to some distribution, this would be a classical EVT setting. One can view the bounds we develop as bounds on a generalization of EVT, where not all samples are created equal.

For general matroid constraints we apply the sampled-mean technique and obtain an approximation comparable to that of applying the greedy algorithm on a monotone submodular function.

**Theorem.** *For any non-negative monotone submodular function there is a deterministic polynomial-time algorithm which optimizes the function under an intersection of $P$ matroids constraint and obtains an approximation ratio arbitrarily close to $1/(1 + P)$ given an approximate oracle.*

**Paper organization.** The main contribution of the paper is the definition of sampled mean approximation in Section 2 and the subsequent analysis of the algorithm for cardinality constraint in Section 3 and matroid constraints in Section 4. The techniques are novel, and the major technical crux of the paper is in analyzing the algorithm in Section 3. Optimization for small rank and lower bounds are in Appendix B.1. In Appendix G we further discuss related work, and in Appendix F we discuss extensions of the algorithms to related models.

## 2 The Sampled Mean Approximation

We begin by defining the sampled mean approximation of a function. This approach considers *bundles* $\mathbf{x}$ of size $c \in \mathcal{O}(1/\epsilon)$. We can then run a variant of a greedy algorithm which adds a bundle of size $c$ in every iteration. For a given bundle $\mathbf{x}$ and set $S$, we define a *ball* to be all bundles obtained by swapping a single element from $\mathbf{x}$ with another not in $S \cup \mathbf{x}$. We denote $\mathbf{x}_{ij} = (\mathbf{x} \setminus \{x_i\}) \cup \{x_j\}$.

**Definition.** *For $S \subset N$ and bundle $\mathbf{x} \subset N$, the ball around $\mathbf{x}$ is $\mathcal{B}_S(\mathbf{x}) := \{\mathbf{x}_{ij} : i \in \mathbf{x}, j \notin S \cup \mathbf{x}\}$.*

We illustrate a ball in Figure 1. Notice that as long as $|S| \leq (1-\delta)|N|$ for some fixed $\delta > 0$, we have that $|\mathcal{B}_S(\mathbf{x})| \in \Omega(n)$. This will allow us to derive weak (but sufficient) concentration bounds.

**Definition.** *Let $f : 2^N \to \mathbb{R}$. For a set $S \subseteq N$ and bundle $\mathbf{x} \subseteq N$, the **mean value**, **noisy mean value**, and **mean marginal contribution** of $\mathbf{x}$ given $S$ are, respectively:*

$$\begin{align}
(1) \qquad F(S \cup \mathbf{x}) &:= \mathbb{E}_{\mathbf{z} \sim \mathcal{B}_S(\mathbf{x})}\big[\; f(S \cup \mathbf{z}) \;\big] \\
(2) \qquad \tilde{F}(S \cup \mathbf{x}) &:= \mathbb{E}_{\mathbf{z} \sim \mathcal{B}_S(\mathbf{x})}\big[\; \tilde{f}(S \cup \mathbf{z}) \;\big] \\
(3) \qquad F_S(\mathbf{x}) &:= \mathbb{E}_{\mathbf{z} \sim \mathcal{B}_S(\mathbf{x})}\big[\; f_S(\mathbf{z}) \;\big]
\end{align}$$

The following lemma implies that under the right conditions, the bundle that maximizes the noisy mean value is a good approximation of the bundle whose (non-noisy) mean value is largest.

**Lemma 2.1.** *Let $\mathbf{x} \in \mathrm{argmax}_{\mathbf{z}:|\mathbf{z}|=c}\, \tilde{F}(S \cup \mathbf{z})$ where $c = 2/\epsilon$. Then, w.p. $\geq 1 - \exp\big(-\Omega\big(n^{1/4}\big)\big)$:*

$$F_S(\mathbf{x}) \geq (1-\epsilon) \max_{\mathbf{z}:|\mathbf{z}|=c} F_S(\mathbf{z}).$$

The above lemma gives us a *weak* concentration bound in the following sense. While it is generally not true that $\tilde{F}(S \cup \mathbf{z}) \approx F(S \cup \mathbf{z})$, we can upper bound $\tilde{F}(S \cup \mathbf{z})$ in a meaningful way and show that $\tilde{F}(S \cup \mathbf{x}^\star) \approx F(S \cup \mathbf{x}^\star)$ for $\mathbf{x}^\star \in \mathrm{argmax}_{\mathbf{z}}\, f_S(\mathbf{z})$ by using submodularity. This allows us to show that the mean marginal contribution of $\mathbf{x}$ that maximizes the *noisy* mean value is an $\epsilon$-approximation to the maximal (non-noisy) mean marginal contribution. Details and proofs are in Appendix A.

In addition to approximating the mean marginal contribution given a noisy oracle, an important property of the sampled-mean approach is that it well-approximates its true marginal contribution.

**Lemma 2.2.** *For any $\epsilon > 0$ and any set $S \subset N$, let $\mathbf{x}$ be a bundle of size $1/\epsilon$, then:*

$$F_S(\mathbf{x}) \geq (1-\epsilon) f_S(\mathbf{x}),$$

The proof is in Appendix A and exploits a natural property of submodular functions: the removal of a random element from a sufficiently large set does not significantly affect its value, in expectation.

Let $\mathbf{x}^\star \in \mathrm{argmax}_{\mathbf{b}:|\mathbf{b}|=c}\, f_S(\mathbf{b})$. Lemma 2.1 and Lemma 2.2 together imply the following corollary:

**Corollary 2.3.** *For a fixed $\epsilon > 0$ let $c = 3/\epsilon$, and $\mathbf{x} \in \mathrm{argmax}_{\mathbf{b}:|\mathbf{b}|=c}\, \tilde{F}(S \cup \mathbf{b})$. Then, w.p. at least $1 - \exp(-\Omega(n^{1/4}))$ we have that: $F_S(\mathbf{x}) \geq (1-\epsilon) f_S(\mathbf{x}^\star)$.*

At a first glance, it may seem as if running the greedy algorithm with $F$ instead of $f$ suffices. The problem, however, is that the *mean* marginal contribution $F_S$ may be an unbounded overestimate of the *true* marginal contribution $f_S$. Consider for example an instance where $S = \emptyset$ and there is a bundle $\mathbf{x}$ of size $c$ s.t. for every $\emptyset \neq \mathbf{z} \subseteq \mathbf{x}$ we have $f(\mathbf{z}) = \delta$ for some arbitrarily small $\delta$, while every other subset $T \subseteq N \setminus \mathbf{x}$ is complementary to $\mathbf{x}$ and has some arbitrarily large value $M$. In this case, $\mathbf{x} = \mathrm{argmax}_{\mathbf{z}:|\mathbf{z}|=c} F(\mathbf{z})$ and $F(\mathbf{x}) = M + \delta$ while $f(\mathbf{x}) = \delta$.

## 3 The Sampled Mean Greedy for Cardinality Constraints

The SM-GREEDY begins with the empty set $S$ and at every iteration considers all bundles (sets) of size $c \in \mathcal{O}(1/\epsilon)$ to add to $S$. At every iteration, the algorithm first identifies the bundle $\mathbf{x}$ which maximizes the noisy mean value. After identifying $\mathbf{x}$, it then considers all possible bundles $\mathbf{z} \in \mathcal{B}_S(\mathbf{x})$ and takes the one whose noisy value is largest. We include a formal description below.

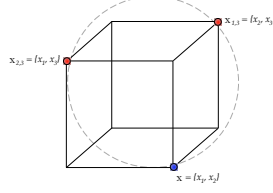

Figure 1: Illustration of a ball around $\mathbf{x} = \{x_1, x_2\}$ where $N = \{x_1, x_2, x_3\}$, and $S = \emptyset$. We think of $\mathbf{x}$ as a point in $[0, 1]^3$ and $\mathcal{B}_S(\mathbf{x}) = \{\mathbf{x}_{12}, \mathbf{x}_{23}\} = \{\{x_2, x_3\}, \{x_1, x_3\}\} = \{(0, 1, 1), (1, 0, 1)\}$.

---

**Algorithm 1** SM-GREEDY

---

**Input:** budget $k$, precision $\epsilon > 0$, $c \leftarrow \frac{56}{\epsilon}$
1: $S \leftarrow \emptyset$
2: **while** $|S| < c \cdot \left\lfloor \frac{k}{c} \right\rfloor$ **do**
3:      $\mathbf{x} \leftarrow \operatorname{argmax}_{\mathbf{b}:|\mathbf{b}|=c} \tilde{F}(S \cup \mathbf{b})$
4:      $\hat{\mathbf{x}} \leftarrow \operatorname{argmax}_{\mathbf{z} \in \mathcal{B}(\mathbf{x})} \tilde{f}(S \cup \mathbf{z})$
5:      $S \leftarrow S \cup \hat{\mathbf{x}}$
6: **end while**
7: **return** $S$

---

A key step in analyzing greedy algorithms like the one above is showing that in every iteration the marginal contribution of the element selected by the algorithm is arbitrarily close to maximal. This can then be used in a standard inductive argument to show that the algorithm obtains an approximation arbitrarily close to $1 - 1/e$. The main crux of the analysis is showing that this property indeed holds in SM-GREEDY w.h.p. when $k \in \Omega(1/\epsilon^2) \cap \mathcal{O}(\sqrt{\log n})$. For $k > \sqrt{\log n}$ the analysis become substantially simpler since it suffices to argue that the algorithm chooses an element whose marginal contribution is approximately optimal in expectation (details are in the proof of Theorem 3.5).

## 3.1   Analysis for $k \in \Omega(1/\epsilon^2) \cap \mathcal{O}(\sqrt{\log n})$

Throughout the rest of this section we will analyze a single iteration of SM-GREEDY in which a set $S \subset N$ was selected in previous iterations and the algorithm adds a bundle $\hat{\mathbf{x}}$ of size $c$. Specifically, $\hat{\mathbf{x}} \in \operatorname{argmax}_{\mathbf{z} \in \mathcal{B}_S(\mathbf{x})} \tilde{f}(S \cup \mathbf{z})$ where $\mathbf{x} \in \operatorname{argmax}_{\mathbf{b}:|\mathbf{b}|=c} \tilde{F}(S \cup \mathbf{b})$.

As discussed above, for $\mathbf{x}^\star \in \operatorname{argmax}_{\mathbf{b}:|\mathbf{b}|=c} f_S(\mathbf{b})$ we want to show that when $c \in \mathcal{O}(1/\epsilon)$:

$$f_S(\hat{\mathbf{x}}) \geq (1 - \epsilon) f_S(\mathbf{x}^\star).$$

To do so, we will define two kinds of bundles in $\mathcal{B}_S(\mathbf{x})$, called **good** and **bad**. A good bundle is a bundle $\mathbf{z}$ for which $f_S(\mathbf{z}) \geq (1 - \frac{2}{3}\epsilon) f_S(\mathbf{x}^\star)$ and a bad bundle $\mathbf{z}$ is one s.t. $f_S(\mathbf{z}) \leq (1 - \epsilon) f_S(\mathbf{x}^\star)$. Our goal is to prove that the bundle $\hat{\mathbf{x}}$ added by the algorithm is w.h.p. not bad. Since according to the definition of good and bad the true marginal contribution of good bundles has a fixed advantage over the true marginal contribution of bad bundles, and $\hat{\mathbf{x}}$ is the bundle with largest *noisy* value, essentially what we need to show is that the largest noise multiplier of a good bundle is sufficiently close to the largest noise multiplier of a bad bundle, with sufficiently high probability.

As a first step we quantify the fraction of good bundles in a ball, which will then allow us to bound the values of the noise multipliers of good and bad bundles. The following claim implies at least half of the bundles in $\mathcal{B}_S(\mathbf{x})$ are good and at most half are bad (the proof is in Appendix B).

**Claim 3.1.** *Suppose $F_S(\mathbf{x}) \geq \left(1 - \frac{\epsilon}{3}\right) f_S(\mathbf{x}^\star)$. Then at least half of the bundles in $\mathcal{B}_S(\mathbf{x})$ are good.*

We next define two thresholds $\xi_{\inf}$ and $\xi_{\sup}$. The threshold $\xi_{\inf}$ is used as a lower bound on the largest value obtained when sampling *at least* $\frac{|B_S(\mathbf{x})|}{2}$ random variables from the noise distribution. Since at least half of the bundles in the ball are good, $\xi_{\inf}$ is a lower bound on the largest noise multiplier of a good bundle. The threshold $\xi_{\sup}$ is used as an upper bound on the largest value obtained when sampling *at most* $\frac{|B_S(\mathbf{x})|}{2}$ random variables from the noise distribution. Since at most half of the bundles in the ball are bad, $\xi_{\sup}$ upper bounds the value of a noise multiplier of a bad bundle. Throughout the rest of this section $\mathcal{D}$ will denote a generalized exponential tail distribution.

**Definition.** *Let* $m = \frac{|\mathcal{B}_S(\mathbf{x})|}{2}$. *For probability density function* $\rho(x)$ *of* $\mathcal{D}$ *we define:*

- $\xi_{\sup}$ *is the value for which:* $\int_{\xi_{\sup}}^{\infty} \rho(x)dx = \frac{2}{m \log n}$;

- $\xi_{\inf}$ *is the value for which:* $\int_{\xi_{\inf}}^{\infty} \rho(x)dx = \frac{2 \log n}{m}$.

**Claim 3.2.** *Let* $m = \frac{|\mathcal{B}_S(\mathbf{x})|}{2}$, $\xi_1, \ldots, \xi_m$ *be i.i.d samples of* $\mathcal{D}$ *and* $\xi^{\star} = \max\{\xi_1, \ldots, \xi_m\}$. *Then:*

$$\Pr \left[ \xi_{\inf} \leq \xi^{\star} \leq \xi_{\sup} \right] \geq 1 - \frac{3}{\log n}$$

The proof is in Appendix B. Since at least half of the bundles in the ball are good and at most half are bad, the above claim implies that with probability $1 - 3/\log n$ the maximal value of noise multipliers of good and bad bundles fall within the range $[\xi_{\inf}, \xi_{\sup}]$. Since this holds in every iteration, when $k \in O(\sqrt{\log n})$ by a union bound we get that this holds in all iterations w.p. $1 - o(1)$.

At this point, our lower bound on the largest noisy value of a good bundle is:

$$\max_{\mathbf{z} \in \text{good}} \tilde{f}(S \cup \mathbf{z}) \geq \xi_{\inf} \times \left( f(S) + \left(1 - \frac{2}{3}\epsilon\right) f_S(\mathbf{x}^{\star}) \right)$$

and the upper bound on the noisy value of any bad bundle is:

$$\max_{\mathbf{z} \in \text{bad}} \tilde{f}(S \cup \mathbf{z}) \leq \xi_{\sup} \times \left( f(S) + (1 - \epsilon) f_S(\mathbf{x}^{\star}) \right).$$

Let $\hat{\mathbf{x}} = \text{argmax}_{\mathbf{z} \in \mathcal{B}_S(\mathbf{x})} \tilde{f}(S \cup \mathbf{z})$. We show that given the right bound on $\xi_{\inf}$ against $\xi_{\sup}$, then $\hat{\mathbf{x}}$ is not a bad bundle (it does not have to be a good bundle). Lemma 3.3 below gives us such a bound.

**Lemma 3.3.** *For any generalized exponential tail distribution, fixed* $\gamma > 0$ *and sufficiently large* $n$:

$$\xi_{\inf} \geq \left(1 - \frac{\gamma}{\sqrt{\log n}}\right) \xi_{\sup}$$

The proof is quite technical and the fully leverages the fact that $k \in \mathcal{O}(\sqrt{\log n})$ and the properties of generalized exponential tail distributions. The main challenge is that the tail of the distribution is not necessarily monotone (see Appendix B for further discussion). We defer the proof to Appendix B.

**Proving the Main Lemma.** We can now prove our main lemma which shows that for any $k \in \omega(1/\epsilon) \cap \mathcal{O}(\sqrt{\log n})$ taking a bundle according to the sample mean approach is guaranteed to be close to the optimal bundle. We state it for a cardinality constraint, but this fact holds more generally for any matroid of rank $k$. We include a short proof sketch and defer the full proof to Appendix B.

**Lemma 3.4.** *Let* $\epsilon > 0$, *and assume that* $k \in \omega(1/\epsilon) \cap \mathcal{O}(\sqrt{\log n})$ *and that* $f_S(\mathbf{x}^{\star}) \in \Omega\left(\frac{f(S)}{k}\right)$. *Then, in every iteration of* SM-GREEDY *we have that with probability at least* $1 - \frac{4}{\log n}$:

$$f_S(\hat{\mathbf{x}}) \geq (1 - \epsilon) f_S(\mathbf{x}^{\star}).$$

***Proof Sketch.*** From Corollary 2.3 we know that in every iteration with overwhelming high probability $F_S(\mathbf{x}) \geq (1 - \delta) f(\mathbf{x}^{\star})$ for $\delta = \epsilon/3$. Since Lemma 3.3 applies for any fixed $\gamma > 0$ we know that for sufficiently large $n$ we can lower bound the value of the maximal good bundle in the ball:

$$\max_{\mathbf{z} \in \text{good}} \tilde{f}(S \cup \mathbf{z}) \geq \left(1 - \frac{\delta^2}{3\sqrt{\log n}}\right) \xi_{\sup} \times [f(S) + (1 - 2\delta) f_S(\mathbf{x}^{\star})]$$

Let $\mathbf{b}$ be the bad bundle with the maximal noisy value for $f_S$. To bound this:

$$\tilde{f}(S \cup \mathbf{b}) = \max_{\mathbf{z} \in \text{bad}} \xi_{S \cup \mathbf{z}} f(S \cup \mathbf{z}) \leq \xi_{\sup} \times [f(S) + (1 - 3\delta) f_S(\mathbf{x}^{\star})]$$

The difference between the two bounds is positive, implying that a bad bundle is not selected. $\square$

**Theorem 3.5.** *For any fixed* $\epsilon > 0$ *and* $k \geq 4/\epsilon^2$ SM-GREEDY *returns a set* $S$ *s.t. w.p.* $1 - o(1)$:

$$f(S) \geq (1 - 1/e - \epsilon) OPT.$$

*Proof.* Let $\kappa = \lfloor \frac{k}{c} \rfloor$ and use $O$ to denote the set of $\kappa$ bundles of size $c$ whose total value is largest. To simplify notation we will treat sets of bundles as sets of elements. We will show that $f(S) \geq (1 - 1/e - \delta)f(O)$ where $\delta = \epsilon/2$. Notice that this implies $f(S) \geq (1 - 1/e - \epsilon)$ since $\kappa > \frac{k}{c} - c$ and by submodularity $f(O) \geq (1 - \delta)\mathrm{OPT}$ when $k \geq \frac{1}{\delta^2} = \frac{4}{\epsilon^2}$.

We introduce some notation: we use $\hat{\mathbf{x}}_i$ to denote the bundle selected at iteration $i \in [\kappa]$, $S_i = \cup_{j \leq i}\hat{\mathbf{x}}_j$, $\mathbf{x}_i \in \mathrm{argmax}_{\mathbf{z}}\,\tilde{F}(S_{i\text{-}1} \cup \mathbf{z})$, $\mathbf{x}_i^\star \in \mathrm{argmax}_{\mathbf{z}}\,f_{S_{i\text{-}1}}(\mathbf{z})$, and $O$ is the set of $\kappa$ bundles with maximal value. Define $\Delta_i = f(O) - f(S_{i\text{-}1})$, and $S_0 = \emptyset$. From submodularity and monotonicity:

$$f_{S_{i\text{-}1}}(\mathbf{x}_i^\star) \geq \frac{1}{\kappa}\Delta_i$$

Consider now a set of bundles $\{\mathbf{z}_1, \ldots, \mathbf{z}_\kappa\}$ where for every $i \in [\kappa]$ we have that $\mathbf{z}_i$ is drawn u.a.r. from $\mathcal{B}_{S_{i\text{-}1}}(\mathbf{x}_i)$. For each such bundle we can assign a random variable $\zeta_i$ for which $f_{S_{i\text{-}1}}(\mathbf{z}_i) = \zeta_i f_{S_{i\text{-}1}}(\mathbf{x}_i^\star)$. Since in every iteration $i \in [\kappa]$ we choose the set whose value is maximal in $\mathcal{B}_{S_{i\text{-}1}}(\mathbf{x}_i)$, by stochastic dominance we know that $f_{S_{i\text{-}1}}(\hat{\mathbf{x}}_i) \geq f(\mathbf{z}_i)$ and therefore:

$$f(S_i) - f(S_{i\text{-}1}) = f_{S_{i\text{-}1}}(\hat{\mathbf{x}}_i) \geq \zeta_i f_{S_{i\text{-}1}}(\mathbf{x}_i^\star) \geq \frac{\zeta_i}{\kappa}\Delta_i$$

We will now show by induction that for all $i \in [\kappa]$ we have that $\Delta_i \leq \prod_{j=1}^{i}\left(1 - \frac{\zeta_j}{\kappa}\right)f(O)$. This is clearly the case for $i = 1$ when $S_0 = \emptyset$ and in general applying the inductive hypothesis we get:

$$\Delta_{i+1} = f(O) - f(S_i) \quad = \Delta_i - (f(S_i) - f(S_{i-1})) \leq \Delta_i\left(1 - \frac{\zeta_{i+1}}{\kappa}\right) \quad \leq \prod_{j=1}^{i+1}\left(1 - \frac{\zeta_j}{\kappa}\right)f(O)$$

We therefore have that:

$$\Delta_\kappa \leq \prod_{j=1}^{\kappa}\left(1 - \frac{\zeta_j}{\kappa}\right)f(O) \leq e^{-\frac{1}{\kappa}\sum_{j=1}^{\kappa}\zeta_j}f(O)$$

Observe that the solution of the algorithm $S$ respects $f(S) = f(S_\kappa) = f(O) - \Delta_\kappa$, thus:

$$f(S) \geq \left(1 - e^{-\frac{1}{\kappa}\sum_{j=1}^{\kappa}\zeta_j}\right)f(O) \tag{1}$$

From Lemma 3.4, when $k \leq \sqrt{\log n}$ for every $i \in [\kappa - 1]$ we have that w.p. $1 - 4/\log(n)$: [3]

$$f_{S_{i\text{-}1}}(\hat{\mathbf{x}}_i) \geq (1 - \delta)f_{S_{i\text{-}1}}(\mathbf{x}_i^\star)$$

Therefore by a union bound, with probability $1 - o(1)$, we have that $\zeta_i \geq (1 - \delta)$ for all $i \in [\kappa]$. In particular, $\frac{1}{\kappa}\sum_{j=1}^{\kappa}\zeta_j \geq (1 - \delta)$. Otherwise, when $k > \sqrt{\log n}$ from Lemma 2.1:

$$\mathbb{E}_{\mathbf{z}\sim\mathcal{B}(\mathbf{x}_i)}[f_{S_{i\text{-}1}}(\mathbf{z})] = F_{S_{i\text{-}1}}(\mathbf{x}_i) \geq \left(1 - \frac{\delta}{2}\right)f_{S_{i\text{-}1}}(\mathbf{x}_i^\star)$$

Thus, $\mathbb{E}[\zeta_{i+1}] \geq (1 - \delta/2)$, and by Chernoff when $\kappa > \sqrt{\log n}$ we get $\frac{1}{\kappa}\sum_{i=1}^{\kappa}\zeta_i \geq (1 - \delta)$ w.p. at least $1 - \exp(-\delta^2\kappa/8)$. Therefore, in both cases, when $k \leq \sqrt{\log n}$ and when $k \geq \sqrt{\log n}$ we have that $\frac{1}{\kappa}\sum_{j=1}^{\kappa}\zeta_j \geq (1 - \delta)$ w.p. $1 - o(1)$. With 1 this implies $f(S) \geq (1 - 1/e - \delta)f(O)$. ☐

**Constant $k$ and information theoretic lower bounds.** For any constant, a single iteration of a minor modification of SM-GREEDY suffices. In Appendix B.1 we show an approximation arbitrarily close to $1 - 1/k$ w.h.p. and $1 - 1/(k+1)$ in expectation. For $k = 1$ this is arbitrarily close to $1/2$. In Appendix D we show nearly matching lower bounds, and in particular that no randomized algorithm can obtain an approximation ratio better than $1/2 + o(1)$ when $k = 1$, and that it is impossible to obtain an approximation better than $(2k - 1)/2k + O(1/\sqrt{n})$ for the optimal set of size $k$.

**Theorem 3.6.** *For any non-negative monotone submodular function there is a deterministic polynomial-time algorithm which optimizes the function under a cardinality constraint $k \geq 3$ and obtains an approximation ratio that is arbitrarily close to $1 - 1/e$ with probability $1 - o(1)$ using access to an approximate oracle. For $k \geq 2$ there is a randomized algorithm whose approximation ratio is arbitrarily close to $1 - 1/e$, in expectation over the randomization of the algorithm. For $k = 1$ the algorithm achieves a $1/2$ approximation in expectation, and no randomized algorithm can achieve an approximation better than $1/2 + o(1)$, in expectation.*

# 4 Approximation Algorithm for Matroids

For intersection of matroids $\mathcal{F}$ the algorithm from the Section 3 is generalized as described below.

---

**Algorithm 2** SM-MATROID-GREEDY

---

**Input:** intersection of matroids $\mathcal{F}$, precision $\epsilon > 0$, $c \leftarrow \frac{56}{\epsilon}$
1: $S \leftarrow \emptyset, X \leftarrow N$
2: **while** $X \neq S$ **do**
3:    $X \leftarrow X \setminus \{\mathbf{x} : S \cup \mathbf{x} \notin \mathcal{F}\}$
4:    $\mathbf{x} \leftarrow \text{argmax}_{\mathbf{b}:|\mathbf{b}|=c} \tilde{F}(S \cup \mathbf{b})$
5:    $\hat{\mathbf{x}} \leftarrow \text{argmax}_{\mathbf{z} \in \mathcal{B}(\mathbf{x})} \tilde{f}(S \cup \mathbf{z})$
6:    $S \leftarrow S \cup \hat{\mathbf{x}}$
7: **end while**
8: **return** $S$

---

The analysis of the algorithm uses the lemma below, which is a generalization of the classic result of [NWF78a]. The proof can be found in the Appendix.

**Lemma 4.1.** *Let $O$ be the optimal solution, $k = |O|$, and for every iteration $i$ of* SM-MATROID-GREEDY *let $S_i$ be the set of elements selected and $\mathbf{x}_i^\star \in \text{argmax}_{|\mathbf{z}|=c} f_{S_{i-1}}(\mathbf{z})$. Then:*

$$f(O) \leq (P+1) \sum_{i=1}^{\frac{k}{c}} f_{S_i}(\mathbf{x}_i^\star)$$

**Theorem 4.2.** *Let $\mathcal{F}$ denote the intersection of $P \geq 1$ matroids on the ground set $N$, and $f : 2^N \to \mathbb{R}$ be a non-negative monotone submodular function. Then with probability $1 - o(1)$ the* SM-MATROID-GREEDY *algorithm returns a set $S \in \mathcal{F}$ s.t.:*

$$f(S) \geq \frac{1-\epsilon}{P+1} OPT$$

*Proof Sketch.* If the rank of the matroid is $\mathcal{O}(1/\epsilon^2)$ we can simply apply the case of small $k$ as in the previous section. Otherwise, assume the rank is at least $\Omega(1/\epsilon^2)$ Let $\kappa = \frac{k}{c}$, $S_i = \{\hat{\mathbf{x}}_1, \ldots, \hat{\mathbf{x}}_i\}$ be current solutions of bundles of size $c$ at iteration $i \in [\kappa]$ of the algorithm, and let $\mathbf{x}_i^\star$ be the optimal bundle at iteration $i$, i.e. $\mathbf{x}_i^\star = \text{argmax}_{\mathbf{b}:|\mathbf{b}|\leq c} f_{S_{i-1}}(\mathbf{b})$ . In every iteration $i \in [\kappa]$, similar to the proof of Theorem 3.5, since we choose the set whose value is maximal in $\mathcal{B}_{S_{i-1}}(\mathbf{x}_i)$ we have:

$$f_{S_{i-1}}(\hat{\mathbf{x}}_i) \geq \zeta_i f_{S_{i-1}}(\mathbf{x}_i^\star)$$

where $\zeta_i$ is a random variable with mean $(1 - \frac{\delta}{2})$. Therefore:

$$f(S) = \sum_{i=1}^{\kappa} f_{i-1}(\hat{\mathbf{x}}_i) \geq \sum_{i=1}^{\kappa} \zeta_i f_{S_{i-1}}(\mathbf{x}_i^\star)$$

From Lemma 3.4, when $k \leq \sqrt{\log n}$ for every $i \in [\kappa - 1]$ we have that w.p. $1 - 4/\log(n)$:

$$f_{S_{i-1}}(\hat{\mathbf{x}}_i) \geq (1-\delta) f_{S_{i-1}}(\mathbf{x}_i^\star)$$

Therefore by a union bound, with probability $1 - o(1)$, we have that $\zeta_i \geq (1 - \delta)$ for all $i \in [\kappa]$. Otherwise, when $k > \sqrt{\log n}$ we apply a Chernoff bound. We get that with probability $1 - o(1)$:

$$f(S) \geq \sum_{i=1}^{\kappa} \zeta_i f_{S_{i-1}}(\mathbf{x}_i^\star) \geq (1-\delta) \sum_{i=1}^{\kappa} f_{S_{i-1}}(\mathbf{x}_i^\star)$$

From Lemma 4.1 this implies the result. $\square$

**Acknowledgements.** A.H. is supported by 1394/16 and by a BSF grant. Y.S. is supported by NSF grant CAREER CCF 1452961, NSF CCF 1301976, BSF grant 2014389, NSF USICCS proposal 1540428, a Google Research award, and a Facebook research award.

## Footnotes

[1]Describing $\tilde{f}$ as a multiplicative approximation of $f$ is more convenient for analyzing multiplicative approximation guarantees. This is w.l.o.g as all our results apply to additive approximations as well.

[2]For example, if for every $S$ the noise is s.t. $\xi_S = 2^{100}$ w.p. $1/2^{100}$ and 0 otherwise, but $n = 50$, it is likely that the oracle will always return 0, in which case we cannot do better than selecting an element at random.

[3] W.l.o.g we assume that in every iteration $f_{S_{i\text{-}1}}(\mathbf{x}_i^\star) \in \Omega(\frac{f(S_{i\text{-}1})}{k})$ to apply Lemma 3.4. Since $\kappa \in \Omega(1/\epsilon)$, $k \in \Omega(1/\epsilon^2)$, and $f(S) \leq \mathrm{OPT}$, ignoring iterations where this does not hold costs $\gamma\epsilon\mathrm{OPT}$ for a small fixed $\gamma$.

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
