[Supplementary Material]

# A    The Sampled Mean Method

Recall that for a bundle $\mathbf{x} = \{x_1, \ldots, x_c\}$ we defined $\mathbf{x}_{ij} = \mathbf{x} \setminus \{x_i\} \cup \{x_j\}$ and the mean value is:

$$F(S \cup \mathbf{x}) = \mathbb{E}_{\mathbf{z} \sim \mathcal{B}_S(\mathbf{x})}[f(\mathbf{z})] = \frac{1}{c} \sum_{i=1}^{c} \left( \frac{1}{t} \sum_{j=1}^{t} f(S \cup \mathbf{x}_{ij}) \right)$$

Throughout this section we make some technical assumptions that essentially hold w.l.o.g:

- First, we assume that $f_S(\mathbf{x}^\star) \in \Omega\left(\frac{f(S)}{k}\right)$. Since we run the algorithm for $\kappa = \lfloor k/c \rfloor$ iterations, ignoring iterations for which this does not hold will cost at most $\gamma \epsilon \mathrm{OPT}$ for some small fixed $\gamma$ of our choice;

- Next, we can assume that $k < (1 - \delta)n$ for some small fixed $\delta$. Otherwise, by selecting $k$ elements u.a.r we obtain an approximation that beats $1 - 1/e$. Since $c \in \mathcal{O}(1/\epsilon)$ we have that $n - |S| - c \in \Omega(n)$ and there are $\Omega(n)$ elements that we can use to swap with elements in $\mathbf{x}$;

- Another assumption we make is that $t \in \omega(n^{1/2}k^2)$ (the choice of $n^{1/2}$ is arbitrary, and we can use $n^{1-\alpha}$ for any fixed $\alpha > 0$). Note that even when $k \in \Omega(n)$ we can make $t \in \Omega(n^d)$ by defining the ball $\mathcal{B}_S(\mathbf{x})$ to be all bundles obtained by removing and swapping $d$ elements from $\mathbf{x}$, instead of swapping a single element as in the current description. Using $d = 3$ for example, suffices. For readability we chose to describe the sampled mean approximation method with a single swap. Since in the worst case we may need to increase the size of the bundles by a factor of 3, we account for this blowup in the description of SM-GREEDY. That is, instead of setting $c = 18/\epsilon$ which suffices when $k \in \mathcal{O}(n^{1/4})$, we use $c = 56/\epsilon$;

- Throughout the entire paper we assume that the values of the distribution are independent of $n$. As discussed in the Introduction, without this assumption no algorithm can obtain any finite approximation. Therefore, since we do not necessarily assume the distribution is bounded, we assume that $n$ is sufficiently large.

**Weak concentration bound.**    We now turn to prove the weak concentration bound. This bound implies that choosing the element that maximizes the noisy sampled mean is an arbitrarily good approximation to choosing the element that maximizes the (non-noisy) sampled mean.

**Lemma.** *2.1 Let* $\mathbf{x} \in \mathrm{argmax}_{\mathbf{z}:|\mathbf{z}|=c} \tilde{F}(S \cup \mathbf{z})$ *where* $c = 2/\epsilon$. *Then, w.p. at least* $1 - \exp\left(-\Omega\left(n^{1/4}\right)\right)$:

$$F_S(\mathbf{x}) \geq (1 - \epsilon) \max_{\mathbf{z}:|\mathbf{z}|=c} F_S(\mathbf{z}).$$

The proof uses lemmas A.1 and A.2. The first lemma lower bounds the noisy mean value of $\mathbf{x}^\star$ against its (non-noisy) mean value, and the second upper bounds the noisy mean value of an arbitrary bundle against its (non-noisy) mean value. We use $\mu$ to denote the mean of the distribution.

**Lemma A.1.** *For a fixed* $\epsilon > 0$, *let* $\mathbf{x}^\star \in \mathrm{argmax}_{\mathbf{z}:|\mathbf{z}|=1/\epsilon} f_S(\mathbf{z})$ *and let* $\eta > 0$ *be a constant. For any* $\lambda > 0$ *we have that with probability at least* $1 - \exp\left(-\Omega(\lambda^2 t^{1/2-\eta})\right)$:

$$\tilde{F}(S \cup \mathbf{x}^\star) \geq (1 - \lambda)\mu \cdot (f(S) + (1 - \epsilon)F_S(\mathbf{x}^\star))$$

*Proof.* Let $\omega = \max_{\mathbf{x}_{ij} \in \mathcal{B}_S(\mathbf{x})} \xi_{ij}$ be the upper bound on values of noise multipliers in the ball, and $t = n - c - |S|$, where $c = 1/\epsilon$. We can break up the sampled mean value to two terms:

$$
\begin{aligned}
\tilde{F}(S \cup \mathbf{x}^\star) &= \frac{1}{c} \sum_{i=1}^{c} \left( \frac{1}{t} \sum_{j=1}^{t} \tilde{f}(S \cup \mathbf{x}_{ij}^\star) \right) \\
&= \frac{1}{c} \sum_{i=1}^{c} \left( \frac{1}{t} \sum_{j=1}^{t} \xi_{ij} f(S \cup \mathbf{x}_{ij}^\star) \right) \\
&= \frac{1}{c} \sum_{i=1}^{c} \left( \frac{1}{t} \sum_{j=1}^{t} \xi_{ij} f(S) + \xi_{ij} f_S(\mathbf{x}_{ij}^\star) \right) \\
&= \frac{1}{c} \sum_{i=1}^{c} \left( \frac{1}{t} \sum_{j=1}^{t} \xi_{ij} f(S) \right) + \frac{1}{c} \sum_{i=1}^{c} \left( \frac{1}{t} \sum_{j=1}^{t} \xi_{ij} f_S(\mathbf{x}_{ij}^\star) \right)
\end{aligned}
$$

For the first term, by a straightforward application of the Chernoff bound we know that it is lower bounded by $(1 - \lambda)\mu f(S)$ with probability at least $1 - \exp(\frac{-\lambda^2 ci}{\omega})$.

To bound the second term, we make the following observations:

- There is at most one set $\mathbf{x}_{-i}^\star$ for which $f_S(\mathbf{x}_{-i}^\star) < \frac{f_S(\mathbf{x}^\star)}{2}$. To see this, assume for purpose of contradiction there are $\mathbf{x}_{-i}^\star$ and $\mathbf{x}_{-j}^\star$ s.t. $f_S(\mathbf{x}_{-i}^\star) \leq f_S(\mathbf{x}_{-j}^\star) < f_S(\mathbf{x}^\star)/2$, then since $\mathbf{x}^\star = \mathbf{x}_{-i}^\star \cup \mathbf{x}_{-j}^\star$, by subadditivity we get a contradiction:
$$
f_S(\mathbf{x}^\star) = f_S(\mathbf{x}_{-i}^\star \cup \mathbf{x}_{-j}^\star) \leq f_S(\mathbf{x}_{-i}^\star) + f_S(\mathbf{x}_{-j}^\star) < 2 \cdot \frac{f_S(\mathbf{x}^\star)}{2} = f_S(\mathbf{x}^\star).
$$

- Let $l$ be in the index of the set with lowest value. Since there is at most one $i \in [c]$ for which $f_S(\mathbf{x}_{-i}^\star) < f_S(\mathbf{x}^\star)$, for all $i \neq l$ we know that the minimal value is at least $f_S(\mathbf{x}^\star)/2$. Due to the maximality of $\mathbf{x}^\star$ we also know that $f_S(\mathbf{x}_{ij}^\star) \leq f_S(\mathbf{x}^\star)$, for all $i \in [c]$. We can therefore apply a Chernoff bound on $\sum_{j=1}^{t} \xi_{ij} f_S(\mathbf{x}_{ij}^\star)$ for every $i \neq l$ and get that w.p. at least $1 - \exp(\frac{\lambda^2 t}{\omega})$:
$$
\frac{1}{t} \sum_{j=1}^{t} \xi_{ij} f_S(\mathbf{x}_{ij}^\star) \geq (1 - \lambda)\mu \frac{1}{t} \sum_{j=1}^{t} f_S(\mathbf{x}_{ij}^\star);
$$

- By the minimality of $l$, and since $c = 1/\epsilon$, we know that:
$$
\frac{1}{c \cdot t} \sum_{i \neq l} \sum_{i=1}^{t} f_S(\mathbf{x}_{ij}^\star) \geq (1 - \epsilon) \frac{1}{c \cdot t} \sum_{i=1}^{c} \sum_{i=1}^{t} f_S(\mathbf{x}_{ij}^\star) = (1 - \epsilon) F_S(\mathbf{x}^\star)
$$

Together, these observations give us our desired bound:

$$
\begin{aligned}
\tilde{F}(S \cup \mathbf{x}^\star) &= \frac{1}{c} \sum_{i=1}^{c} \left( \frac{1}{t} \sum_{j=1}^{t} \xi_{ij} f(S) \right) + \frac{1}{c} \sum_{i=1}^{c} \left( \frac{1}{t} \sum_{j=1}^{t} \xi_{ij} f_S(\mathbf{x}_{ij}^\star) \right) \\
&\geq \frac{1}{c} \sum_{i=1}^{c} \left( \frac{1}{t} \sum_{j=1}^{t} \xi_{ij} f(S) \right) + \frac{1}{c} \sum_{i \neq l} \left( \frac{1}{t} \sum_{j=1}^{t} \xi_{ij} f_S(\mathbf{x}_{ij}^\star) \right) \\
&\geq (1 - \lambda)\mu f(S) + (1 - \lambda)\mu \frac{1}{c} \sum_{i \neq l} \sum_{j=1}^{t} f_S(\mathbf{x}_{ij}^\star) \\
&\geq (1 - \lambda)\mu f(S) + (1 - \lambda)\mu(1 - \epsilon) F_S(\mathbf{x}^\star) \\
&= (1 - \lambda)\mu \left( f(S) + (1 - \epsilon) F_S(\mathbf{x}^\star) \right)
\end{aligned}
$$

Finally, to upper bound $\omega$, since the noise distribution is a generalized exponential tail we have that for any $\delta > 0$ and sufficiently large $m$:

$$\Pr[\omega < m^\delta] > 1 - \exp\left(-\Omega\left(\frac{m^\delta}{\log m}\right)\right)$$

To see this, note that this is trivially true when $m$ tends to infinity when the noise distribution is bounded, or has finite support. If the noise distribution is unbounded, since its tail is subexponential, at any given sample the probability of seeing the value $m^\delta$ is at most $\exp(-\mathcal{O}(m^\delta))$ and iterating this a polynomial number of times achieves the bound.

Therefore we know that with probability $1 - \exp\left(-\Omega(\sqrt{c \cdot t}\log^{-1}(c \cdot t))\right)$ all $c \cdot t = |\mathcal{B}_S(\mathbf{x})|$ noise multipliers are bounded from above by $\omega = \sqrt{ct}$. Thus by a union bound, the bound above holds with probability of at least $1 - \exp\left(-\Omega(\lambda^2 t^{1/2-\eta})\right)$ for some arbitrarily small $\eta > 0$. $\qquad\square$

The following bound shows that for sufficiently large $t$ (which is proportional to $n$), we have that $\widetilde{F}(S) \approx (1+\lambda)\mu(F(S) + 3t^{-1/4}f_S(\mathbf{x}^\star))$ for small $\lambda > 0$ .

**Lemma A.2.** *For $\epsilon > 0$, let $\mathbf{x}$ be a bundle of size $c = 1/\epsilon$ and let $\eta > 0$ be a constant. For any $\lambda > 0$ we have that with probability at least $1 - \exp\left(-\Omega\left(\lambda^2 t^{1/2-\eta}\right)\right)$:*

$$\widetilde{F}(S \cup \mathbf{x}) < (1+\lambda)\mu \cdot \left(f(S) + F_S(\mathbf{x}) + 3t^{-\eta/3}f_S(\mathbf{x}^\star)\right).$$

*Proof.* As in the proof of the previous lemma, let $\omega$ be the upper bound on the value of a noise multipliers. Since $\tilde{F}(\mathbf{x})$ is an average of these values over all $i \in [c]$, a concentration bound that holds for every $i \in [c]$ will give us the desired result.

For a given $i \in [c]$, let $\mathbf{z}_1, \ldots, \mathbf{z}_t$ denote the bundles $\mathbf{x}_{i1}, \ldots, \mathbf{x}_{it}$. For each bundle $\mathbf{z}_i$ we will use $\alpha_i$ to denote the marginal value $f_S(\mathbf{z}_i)$ and $\xi_i$ to denote the noise multiplier $\xi_{S \cup \mathbf{z}_i}$. We have two sums:

$$\frac{1}{t}\sum_{i=1}^{t}\tilde{f}(S \cup \mathbf{x}_{ij}) = \frac{1}{t}\sum_{i=1}^{t}\xi_i f(S) + \frac{1}{t}\sum_{i=1}^{t}\xi_i\alpha_i. \tag{2}$$

As before, we can immediately apply a Chernoff bound on the first term and what remains is to show concentration on the second term. Define $\alpha^\star = \max_i \alpha_i$. Note that due to the maximality of $\mathbf{x}^\star$ we have that $\alpha^\star \leq f_S(\mathbf{x}^\star)$. To apply concentration bounds on the second term, we partition the values of $\{\alpha_i\}_{i\in[t]}$ to bins of width $\alpha^\star \cdot t^{-\nu}$ for some arbitrarily small constant $\nu > 0$. We call a bin *dense* if it has at least $t^{1-2\nu}$ values and *sparse* otherwise. Using this terminology:

$$\sum_{i=1}^{t}\xi_i\alpha_i = \sum_{i\in\text{dense}}\xi_i\alpha_i + \sum_{i\in\text{sparse}}\xi_i\alpha_i.$$

Let $\text{BIN}_\ell$ be the dense bin whose elements have the largest values. Consider the $t^{1-2\nu}/2$ largest values in $\text{BIN}_\ell$ and call the set of indices associated with these values $L$. We can rewrite the sum as:

$$\sum_{i=1}^{t}\xi_i\alpha_i = \sum_{i\in\text{dense}\setminus L}\xi_i\alpha_i + \sum_{i\in L\cup\text{sparse}}\xi_i\alpha_i$$

The set $L \cup \text{sparse}$ is of size at least $t^{1-2\nu}/2$ and at most $t^{1-2\nu}/2 + t^{1-\nu}$. This is because $L$ is of size exactly $t^{1-2\nu}/2$ and there are at most $t^{1-\nu}$ values in bins that are sparse since there are $t^\nu$ bins, and sparse bins have less than $t^{1-2\nu}$ values. Thus, when $\omega$ is an upper bound on the value of the noise multiplier, from Chernoff, for any $\lambda < 1$ with probability $1 - \exp(-\Omega(\lambda^2 t^{1-2\nu}\omega^{-1}))$:

$$\sum_{i\in L\cup\text{sparse}}\xi_i\alpha_i \leq \sum_{i\in L\cup\text{sparse}}\xi_i\alpha^\star$$
$$< (1+\lambda)\mu \cdot |L \cup \text{sparse}| \cdot \alpha^\star$$
$$\leq (1+\lambda)\mu \cdot \left(\frac{t^{1-2\nu}}{2} + t^{1-\nu}\right)\alpha^\star$$
$$< (1+\lambda)\mu \cdot 2t^{1-\nu}\alpha^\star$$

We will now apply concentration bounds on the values in the dense bins. For a dense bin $\text{BIN}_q$, let $\alpha_{\max(q)}$ and $\alpha_{\min(q)}$ be the maximal and minimal values in the bin, respectively. Since the width of the bins is $\alpha^\star \cdot t^{-\nu}$ we have that $\alpha_{\min(q)} \geq \alpha_{\max(q)} - \alpha^\star \cdot t^{-\nu}$. Recall that a dense bin has at least $t^{1-2\nu}$ values. We can therefore apply a Chernoff bound on a dense bin $\text{BIN}_q$. We get that for any $\lambda < 1$ w.p. $1 - \exp(-\lambda^2 t^{1-2\nu}\omega^{-1})$:

$$\sum_{i \in \text{BIN}_q} \xi_i \alpha_i \leq \sum_{i \in \text{BIN}_q} \xi_i \cdot \alpha_{\max(q)}$$
$$\leq (1+\lambda)\mu \cdot \alpha_{\max(q)} \cdot |\text{BIN}_q|$$
$$\leq (1+\lambda)\mu \cdot \left(\alpha_{\min(q)} + \alpha^\star \cdot t^{-\nu}\right) \cdot |\text{BIN}_q|$$
$$< (1+\lambda)\mu \cdot \left(|\text{BIN}_q| \cdot \alpha_{\min(q)} + |\text{BIN}_q|\alpha^\star \cdot t^{-\nu}\right)$$

Applying a union bound over all bins we get with probability $1 - t^\nu \cdot \exp(-\lambda^2 t^{1-2\nu}\omega^{-1})$:

$$\sum_{i \in \text{dense}\backslash L} \xi_i \alpha_i < \sum_q (1+\lambda)\mu \cdot \left(|\text{BIN}_q|\alpha^\star \cdot t^{-\nu} + |\text{BIN}_q| \cdot \alpha_{\min(q)}\right) < (1+\lambda)\mu \cdot \left(\alpha^\star t^{1-\nu} + \sum_{i=1}^{t} \alpha_i\right)$$

Conditioning on both events, together we have:

$$\frac{1}{t}\sum_{j=1}^{t} f_S(\mathbf{x}_{ij}) = \frac{1}{t}\sum_{i=1}^{t} \xi_i \alpha_i = \frac{1}{t}\left(\sum_{i \in L \cup \text{sparse}} \xi_i \alpha_i + \sum_{i \in \text{dense}\backslash L} \xi_i \alpha_i\right) < (1+\lambda)\mu \cdot \left(3t^{-\nu} f_S(\mathbf{x}^\star) + F_S(\mathbf{x})\right)$$

By a union bound over all $i \in [c]$ we get that with probability $1 - \exp(-\Omega(\lambda^2 t^{1-3\nu}\omega^{-1}))$:

$$\tilde{F}(S\cup\mathbf{x}) = \frac{1}{c}\sum_{i=1}^{c}\left(\frac{1}{t}\sum_{j=1}^{t}\xi_{ij}f(S) + \frac{1}{t}\sum_{j=1}^{t}\xi_{ij}f_S(\mathbf{x}_{ij})\right) \leq (1+\lambda)\mu\left(f(S) + F_S(\mathbf{x}) + 3t^{-\nu}f_S(\mathbf{x}^\star)\right)$$

As in the previous lemma, since the distribution is a generalized exponential tail, we know that $\omega \leq \sqrt{c \cdot t}$ w.p. at least $1 - \exp\left(-\Omega((c \cdot t)^{1/2}\log^{-1}(c \cdot t))\right)$. Taking a union bound, the bound above holds with probability of at least $1 - \exp\left(-\Omega(\lambda^2 t^{1/2-3\nu})\right)$. Setting $\eta = \nu/3$ concludes our proof. $\square$

***Proof of Lemma 2.1***. Let $\mathbf{x}^\star = \arg\max_{\mathbf{x}:|\mathbf{x}|=c} f_S(\mathbf{x})$ and let $\mathbf{b}$ be a bundle of size $c$ for which $F_S(\mathbf{b}) < (1-\epsilon)F_S(\mathbf{x}^\star)$. We will show that $\mathbf{b}$ cannot be selected as $\mathbf{x}^\star$ beats $\mathbf{b}$ with overwhelming high probability:

$$\tilde{F}(S \cup \mathbf{x}^\star) > \tilde{F}(S \cup \mathbf{b}).$$

By taking a union bound over all possible $\mathcal{O}(n^c)$ bundles we will then conclude that the bundle whose noisy mean contribution is largest must have mean contribution at least factor of $(1-\epsilon)$ from that of $\mathbf{x}^\star$, with overwhelming high probability.

From Lemma A.1 since $c = 2/\epsilon$ we know that for $\lambda \in [0,1)$ with probability $1 - \exp\left(-\Omega(\lambda^2 t^{1/2-\eta})\right)$:

$$F(S \cup \mathbf{x}^\star) \geq (1-\lambda)\mu\left(f(S) + \left(1 - \frac{\epsilon}{2}\right)F_S(\mathbf{x}^\star)\right)$$

Similarly, from Lemma A.2 we know that with probability $1 - \exp\left(-\Omega(\lambda^2 t^{1/2-\eta})\right)$:

$$F(S \cup \mathbf{b}) \geq (1+\lambda)\mu\left(f(S) + F_S(\mathbf{b}) + 3t^{-\eta/3}f_S(\mathbf{x}^\star)\right)$$
$$\geq (1+\lambda)\mu\left(f(S) + (1 - \epsilon + 6t^{-\eta/3})F_S(\mathbf{x}^\star)\right)$$

where we used the assumption $F_S(\mathbf{x}^\star) \geq (1-\epsilon)F_S(\mathbf{b})$ and the fact that $2F_S(\mathbf{x}^\star) \geq f_S(\mathbf{x}^\star)$. Recall that we're assuming that $f_S(\mathbf{x}^\star) \in \Omega\left(\frac{f(S)}{k}\right)$. Thus, for some small fixed $\alpha > 0$

we have that $F_S(\mathbf{x}^\star) \geq \alpha f(S)/2k$. Given the inequalities above, with probability at least $1 - \exp\left(-\Omega(\lambda^2 t^{1/2-\eta})\right)$:

$$\tilde{F}(S \cup \mathbf{x}^\star) - \tilde{F}(S \cup \mathbf{b}) \geq \mu\left(\left(\frac{\epsilon}{2} - 2\lambda - (1+\lambda)6t^{-\eta/3}\right)F_S(\mathbf{x}^\star) - 2\lambda f(S)\right)$$

$$\geq \mu\left(\left(\frac{\epsilon}{2} - 2\lambda - 12t^{-\eta/3} - \lambda\frac{k}{\alpha}\right)F_S(\mathbf{x}^\star)\right)$$

For any $\lambda \leq \epsilon^2/4k$ the difference above is strictly positive. Without loss of generality we're assuming $t \gg n^{1/2} \cdot k^2$ and therefore the difference is positive with probability $1 - \exp\left(-\Omega(t^{1/4})\right)$. Since $t \in \Omega(n)$ this concludes our proof. $\qquad \square$

**The sampled mean is (almost) an upper bound of the function.** We now show that when the size of the bundle is sufficiently large, the marginal contributions of the sampled mean nearly upper bound the true marginal contributions of a monotone submodular function.

**Lemma.** *2.2 For any $\epsilon > 0$ and any set $S \subset N$, let $\mathbf{x}$ be a bundle of size $1/\epsilon$, then:*

$$F_S(\mathbf{x}) \geq (1 - \epsilon)f_S(\mathbf{x})$$

*Proof.* Let $c = 1/\epsilon$ and consider an arbitrary ordering on the elements in the bundle $x_1, \ldots, x_c \in \mathbf{x}$. Define $\mathbf{x}_{-i} = \mathbf{x} \setminus \{x_i\}$, and $\mathbf{x}_{ij} = \mathbf{x}_{-i} \cup \{x_j\}$. From submodulairty we get that for any $i \in [c]$:

$$f_{S \cup \mathbf{x}_{-i}}(x_i) = f(S \cup \mathbf{x}_{-i} \cup x_i) - f(S \cup \mathbf{x}_{-i}) \leq f(S \cup \{x_1 \ldots, x_i\}) - f(S \cup \{x_1, \ldots, x_{i-1}\})$$

Thus:

$$\sum_{i=1}^{c} f_{S \cup \mathbf{x}_{-i}}(x_i) \leq \sum_{i=1}^{c} (f(S \cup \{x_1 \ldots, x_i\}) - f(S \cup \{x_1, \ldots, x_{i-1}\})) = f_S(\mathbf{x}) \qquad (3)$$

Let $t = n - c - |S|$. By summing over all $\mathbf{x}_{-i}$ we get the desired bound:

$$\begin{aligned}
F_S(\mathbf{x}) &= \frac{1}{c \cdot t} \sum_{j=1}^{t} \sum_{i=1}^{c} f_S(\mathbf{x}_{ij}) \\
&\geq \frac{1}{c} \sum_{i=1}^{c} f_S(\mathbf{x}_{-i}) \\
&= \frac{1}{c} \sum_{i=1}^{c} (f_S(\mathbf{x}_{-i} \cup x_i) - f_{S \cup \mathbf{x}_{-i}}(x_i)) \\
&= \frac{1}{c} \sum_{i=1}^{c} f_S(\mathbf{x}) - \frac{1}{c} \sum_{i=1}^{c} f_{S \cup \mathbf{x}_{-i}}(x_i) \\
&\geq f_S(\mathbf{x}) - \frac{1}{c} f_S(\mathbf{x}) \qquad \qquad \text{by (3)} \\
&= \left(1 - \frac{1}{c}\right) f_S(\mathbf{x}) \\
&= (1 - \epsilon) f_S(\mathbf{x}).
\end{aligned}$$

$\qquad \square$

# B  The Sampled-Mean Greedy

**Claim.** *3.1. Suppose* $F(\mathbf{x}) \geq \left(1 - \frac{\epsilon}{3}\right) f_S(\mathbf{x}^\star)$. *Then at least half of the bundles in* $\mathcal{B}_S(\mathbf{x})$ *are good.*

*Proof.* For convenience we will use $\delta = \epsilon/3$. Let $\mathcal{B}_S^+(\mathbf{x})$ be the set of good bundles in $\mathcal{B}_S(\mathbf{x})$. Due to the maximality of $\mathbf{x}^\star$ we have that $f_S(\mathbf{z}) \leq f_S(\mathbf{x}^\star)$ for every $\mathbf{z} \in \mathcal{B}_S(\mathbf{x})$. Therefore:

$$\sum_{\mathbf{z} \in \mathcal{B}_S(\mathbf{x})} f_S(\mathbf{z}) \leq |\mathcal{B}_S^+(\mathbf{x})| \cdot f_S(\mathbf{x}^\star) + \left(|\mathcal{B}_S(\mathbf{x})| - |\mathcal{B}_S^+(\mathbf{x})|\right) \cdot (1 - 2\delta) f_S(\mathbf{x}^\star) \tag{4}$$

By the definition of $F(\mathbf{x}) = \mathbb{E}_{\mathbf{z} \in \mathcal{B}_S(\mathbf{x})}[f(\mathbf{z})]$ and our assumption $F(\mathbf{x}) \geq (1 - \delta) f(\mathbf{x})$:

$$\frac{1}{|\mathcal{B}_S(\mathbf{x})|} \sum_{\mathbf{z} \in \mathcal{B}_S(\mathbf{x})} f_S(\mathbf{z}) \geq (1 - \delta) f_S(\mathbf{x}^\star) \tag{5}$$

Putting (4) and (5) together we get:

$$|\mathcal{B}_S(\mathbf{x})|(1 - \delta) f_S(\mathbf{x}^\star) \leq \left(|\mathcal{B}_S^+(\mathbf{x})| + (|\mathcal{B}_S(\mathbf{x})| - |\mathcal{B}_S^+(\mathbf{x})|)(1 - 2\delta)\right) f_S(\mathbf{x}^\star)$$

Rearranging we get that $|\mathcal{B}_S^+(\mathbf{x})| \geq |\mathcal{B}_S(\mathbf{x})|/2$, as required. $\qquad\square$

**Claim.** *3.2 Let* $m = \frac{|\mathcal{B}_S(\mathbf{x})|}{2}$, $\xi_1, \ldots, \xi_m$ *be i.i.d samples from* $\mathcal{D}$ *and* $\xi^\star = \max\{\xi_1, \ldots, \xi_m\}$. *Then:*

$$\Pr\left[\xi_{\inf} \leq \xi^\star \leq \xi_{\sup}\right] \geq 1 - \frac{3}{\log n}$$

*Proof.* For a single sample $\xi$ from $\mathcal{D}$, we have that $\Pr[\xi \leq \xi_{\sup}] = 1 - \frac{2}{m \log n}$. If we take $m$ independent samples $\xi_1, \ldots \xi_m$, the probability they are all bounded by $\xi_{\sup}$ is:

$$\left(1 - \frac{2}{m \log n}\right)^m > \left(1 - \frac{2}{\log n}\right)$$

For $\xi_{\inf}$, the probability a single sample $\xi$ taken from $\mathcal{D}$ is at most $\xi_{\inf}$ is $\Pr[\xi \leq \xi_{\inf}] = 1 - \frac{2 \log n}{m}$. If we take independent samples $\xi_1, \ldots \xi_m$, the probability they are all bounded by $\xi_{\inf}$ is:

$$\left(1 - \frac{2 \log n}{m}\right)^m < \frac{2}{2^{\log n}} = \frac{2}{n}$$

Therefore, by a union bound the likelihood that the maximum of $m$ samples is bounded between $\xi_{\inf}$ and $\xi_{\sup}$ is at least $1 - \frac{3}{\log n}$. $\qquad\square$

**Bounding extreme values of the noise multipliers.** Before proving Lemma 3.3, we illustrate the main challenge. First, consider a distribution which returns 0 with probability 0.99 and 1 with probability 0.01. If $m = 50$, clearly the lemma doesn't hold, but for $n > 1000$ the lemma would follow through. It is easy to generalize the problem to any distribution with an atom at its supremum.

One class of distributions for which the lemma may not hold, is one with an infinite number of atoms. For example, consider the distribution for which $\Pr[2^d] = 1/2^d$. In this case, the lemma is incorrect regardless of the value of $m$. The problem is not with the atoms, as it is easy to construct a density function which is non zero only around $2^d$, and its integral around $2^d$ is exactly $2^{-d}$. Note however that such a density function would be far from monotone. We do not want to require monotone noise distributions, as to not rule out bimodal distributions, and to allow for small fluctuations in the density function. Instead, we require that except for a finite number of modalities, the function's tail has a lower bound and an upper bound, which are somehow related. This requirement is rather weak, and encompasses in particular Exponential distributions, Gaussians (which are monotone), bounded distributions and distributions with a finite support.

**Lemma.** *3.3 For any generalized exponential tail distribution, fixed* $\gamma > 0$ *and sufficiently large* $n$:

$$\xi_{\inf} \geq \left(1 - \frac{\gamma}{\sqrt{\log n}}\right) \xi_{\sup}$$

*Proof.* We will use $\varphi = \frac{\gamma}{\sqrt{\log n}}$ and first we give a proof for distributions with bounded support. Let $M$ be an upper bound on $\mathcal{D}$. If there is an atom at $M$ with some probability $p > 0$, then we are done, as $\xi_{\text{inf}} = \xi_{\text{sup}} = M$. Otherwise, since $\mathcal{D}$ has a generalized exponential tail we know that $\rho(M) = p$ for some $p > 0$, and that $\rho$ is continuous at $M$. But then there is some $\delta > 0$ such that for any $M - \delta \leq x \leq M$ we have that $\rho(x) \geq p/2$. Choosing $n$ to be large enough that $(1 - \epsilon)p > p - \delta$:

$$\int_{(1-\epsilon)M}^{M} \rho(x) \geq p/2\epsilon$$

Choosing $n$ large enough s.t. $2 \log n / m < \gamma/2\epsilon$ gives that $\xi_{\text{inf}} \geq (1 - \epsilon)M$. As $\xi_{\text{sup}} \leq M$ we are done.

When the distribution does not have bounded support, recall that by definition of $\mathcal{D}$ for $x \geq x_0$, we have that $\rho(x) = e^{-g(x)}$, where $g(x) = \sum_i a_i x_i^{\alpha}$ and that we do not assume that all the $\alpha_i$'s are integers, but only that $\alpha_0 \geq \alpha_1 \geq \ldots$, and that $\alpha_0 \geq 1$. We do not assume anything on the other $\alpha_i$ values. In this case, the proof follows three stages:

1. We use properties of $\mathcal{D}$ to argue upper and lower bounds for $\rho(x)$;

2. We show an upper bound $M$ on $\xi_{\text{sup}}$;

3. We show that integrating a lower bound of $\rho(x)$ from $(1 - \varphi)M$ to $\infty$, yields a probability mass at least $\frac{\log n}{\varphi m}$. Now suppose for contradiction that $\xi_{\text{inf}} < (1 - \varphi)\xi_{\text{sup}}$, we would get that $\int_{\xi_{\text{inf}}}^{\infty} \rho(x)$ is strictly greater than $\frac{\log n}{\varphi m}$, which contradicts the definition of $\xi_{\text{inf}}$.

We now elaborate each on stage.

**First stage.** For the first stage we will show that for every $g(x)$, there exists $n_0$ such that for any $n > n_0$ and $x \geq \left(\frac{\log n}{2a_0}\right)^{1/\alpha_0}$ we have that for $\beta = \varphi/100 < 1/100$:

$$(1 + \beta)a_0 x^{\alpha_0 - 1} e^{-(1+\beta)a_0 x^{\alpha_0}} \leq \rho(x) \leq (1 - \beta)a_0 x^{\alpha_0 - 1} e^{-(1-\beta)a_0 x^{\alpha_0}}$$

We explain both directions of the inequality. To see $a_0 x^{\alpha_0 - 1}(1 + \beta)e^{-(1+\beta)a_0 x^{\alpha_0}} \leq \rho(x)$ we first show:

$$e^{-(1+\beta/2)a_0 x^{\alpha_0}} \leq \rho(x)$$

This holds since for sufficiently large $n$, we have that:

$$x \geq \frac{(\log n)^{1/\alpha_0}}{2a_0} \geq \left(\frac{2\sum_{i=1} |a_i|}{\beta a_0}\right)^{\alpha_0 - \alpha_1}$$

So the term $\frac{\beta}{2} x^{\alpha_0}$ dominates the rest of the terms. We now show that:

$$e^{-(1+\beta/2)a_0 x^{\alpha_0}} \geq a_0 x^{\alpha_0 - 1}(1 + \beta)e^{-(1+\beta)a_0 x^{\alpha_0}}$$

This is equivalent to:

$$e^{\beta a_0 / 2 x^{\alpha_0}} \geq a_0 x^{\alpha_0 - 1}(1 + \beta)$$

Which hold for $x = \log \log^3 n$ and large enough $n$.

The other side of the inequality is proved in a similar way. We want to show that:

$$\rho(x) \leq (1 - \beta)a_0 x^{\alpha_0 - 1} e^{-(1-\beta)a_0 x^{\alpha_0}}$$

Clearly for $x > \log \log^3 n$ we have that $(1 - \beta)a_0 x^{\alpha_0 - 1} > 1$. Hence we just need to show that:

$$\rho(x) \leq e^{-(1-\beta)a_0 x^{\alpha_0}}$$

But this holds for sufficiently large $n$ s.t.:

$$x \geq \frac{(\log n)^{1/\alpha_0}}{2a_0} \geq \left(\frac{\sum_{i=1} |a_i|}{\beta a_0}\right)^{\alpha_0 - \alpha_1}$$

**Second stage.** We now proceed to the second stage, and compute an upper bound on $\xi_{\sup}$. Note that if for every $x \geq M$ we have $\rho(x) \leq g(x)$ and

$$\int_{\xi_{\sup}}^{\infty} \rho(x) = \int_{M}^{\infty} g(x)$$

then it must be that $M \geq \xi_{\sup}$. Applying this to our setting, and using $m = \mathcal{B}_S(\mathbf{x}) = c \cdot (n - |S| - c)$ we bound $\rho(x) \leq (1-\beta)a_0 x^{\alpha_0 - 1} e^{-(1-\beta)a_0 x^{\alpha_0}}$ to get:

$$\frac{1}{m \log n} = \int_{M}^{\infty} (1-\beta)a_0 x^{\alpha_0 - 1} e^{-(1-\beta)a_0 x^{\alpha_0}} = -e^{-(1-\beta)a_0 x^{\alpha_0}} |_{M}^{\infty} = e^{-(1-\beta)a_0 M^{\alpha_0}}$$

Taking the logarithm of both sides, we get:

$$-(1-\beta)a_0 M^{\alpha_0} = \log \frac{1}{m \log n} = -\log(m \log n)$$

Multiplying by $-1$, dividing by $(1-\beta)a_0$ and taking the $1/\alpha_0$ root we get:

$$M = \left( \frac{\log m \log n)}{(1-\beta)a_0} \right)^{\alpha_0}$$

Note that $(1 - \varphi)M > \left( \frac{\log n}{2a_0} \right)^{1/\alpha_0}$ and hence our bounds on $\rho(x)$ hold for this regime.

**Third stage.** We move to the third stage, and bound $\int_{(1-\varphi)M}^{\infty} \rho(x)$ from below. If we show that: $\int_{(1-\varphi)M}^{\infty} \rho(x)$ is greater than $\frac{\log n}{\varphi m}$, this implies that $\xi_{\inf} \geq (1 - \varphi)M$, as $\xi_{\inf}$ is defined as the value such that when we integrate $\rho(x)$ from $\xi_{\inf}$ to $\infty$ we get exactly $\frac{\log n}{\varphi m}$. We show:

$$\int_{(1-\varphi)M}^{\infty} \rho(x) \geq (1+\beta)a_0\alpha_0 x^{\alpha_0 - 1} e^{-(1+\beta)a_0 x^{\alpha_0}}$$

$$= -e^{-(1+\beta)a_0 x^{\alpha_0}} |_{(1-\varphi)M}^{\infty}$$

$$= e^{-(1+\beta)a_0((1-\varphi)M)^{\alpha_0}}$$

$$= e^{-(1+\beta)a_0 M^{\alpha_0}(1-\varphi)^{\alpha_0}}$$

$$\geq e^{-(1+\beta)a_0 M^{\alpha_0}(1-\varphi)}$$

However $a_0 M^{\alpha_0} = \left( \frac{\log m \log n)}{(1-\beta)} \right)$. Since $\beta < 0.1$ we have that $\frac{1+\beta}{1-\beta} < 1 + 3\beta$. Substituting both expressions we get:

$$e^{-(1+\beta)a_0 M^{\alpha_0}(1-\varphi)} \geq e^{-(1+3\beta)(1-\varphi)\log m \log n)}$$

$$= \left( \frac{1}{m \log n} \right)^{(1-\varphi)(1+3\beta)}$$

$$\geq \left( \frac{1}{m \log n} \right)^{(1-\varphi/2)}$$

where we used that $\beta = \varphi /100$ and hence $(1 - \varphi)(1 + 3\beta) < 1 - \varphi /2$. We now need to compare this to $\frac{\sqrt{\log n}}{\varphi m}$. To do this, note that:

$$\left( \frac{1}{m \log n} \right)^{(1-\varphi/2)} \geq \frac{1}{m^{1-\varphi/2} \log n} \geq \frac{2^{\sqrt{\log n}}}{m \log n} \geq \frac{\log n}{\varphi m}$$

where $n$ is large enough that $\frac{\varphi}{2} \log m > \sqrt{\log n}$. This completes the proof, since $\xi_{\inf} \geq (1 - \varphi)M \geq (1 - \varphi)\xi_{\sup}$ as required. $\qquad\square$

**Lemma.** *3.4 Let $\epsilon > 0$, and assume that $k \in \omega(1/\epsilon) \cap \mathcal{O}(\sqrt{\log n})$ and that $f_S(\mathbf{x}^{\star}) \in \Omega\left( \frac{f(S)}{k} \right)$. Then, in every iteration of* SM-GREEDY *we have that with probability at least $1 - \frac{4}{\log n}$:*

$$f_S(\hat{\mathbf{x}}) \geq (1 - \epsilon)f_S(\mathbf{x}^{\star}).$$

*Proof.* From Corollary 2.3 we know that when $c = 9/\epsilon$, then in every iteration with overwhelming high probability we have that for $\delta = \epsilon/3$:

$$F_S(\mathbf{x}) \geq (1 - \delta)f(\mathbf{x}^\star) \tag{6}$$

In every iteration the algorithm selects $\hat{\mathbf{x}} \in \operatorname{argmax}_{\mathbf{z} \in \mathcal{B}_S(\mathbf{x})} \tilde{f}(S \cup \mathbf{z})$. Recall that a good bundle is $\mathbf{z} \in \mathcal{B}_S(\mathbf{x})$ for which $f(\mathbf{z}) \geq (1 - 2\epsilon/3)f(\mathbf{x}^\star)$ and a bad bundle is $\mathbf{z} \in \mathcal{B}_S(\mathbf{x})$ s.t. $f(\mathbf{z}) \leq (1 - \epsilon)f(\mathbf{x}^\star)$. Conditioning on the assumption (6), from Claim 3.2 we know that with probability at least $1 - \frac{3}{\log n}$ the noise multipliers of both good and bad bundles in $\mathcal{B}_S(\mathbf{x})$ are in $[\xi_{\inf}, \xi_{\sup}]$. Since Lemma 3.3 applies for any fixed $\gamma > 0$ we know that for sufficiently large $n$ we have that:

$$\xi_{\inf} \geq \left(1 - \frac{\delta^2}{3\sqrt{\log n}}\right)\xi_{\sup}$$

Thus, a lower bound on the maximal noisy value of a *good* bundle is:

$$
\begin{aligned}
\max_{\mathbf{z} \in \text{good}} \tilde{f}(S \cup \mathbf{z}) &= \max_{\mathbf{z} \in \text{good}} \xi_{\mathbf{z}} \times [f(S) + f_S(\mathbf{z})] \\
&\geq \xi_{\inf} \times [f(S) + (1 - 2\delta)f_S(\mathbf{x}^\star)] \\
&\geq \left(1 - \frac{\delta^2}{3\sqrt{\log n}}\right)\xi_{\sup} \times [f(S) + (1 - 2\delta)f_S(\mathbf{x}^\star)]
\end{aligned}
$$

An upper bound on the maximal noisy value of a *bad* bundle is:

$$\tilde{f}(S \cup \mathbf{b}) = \max_{\mathbf{z} \in \text{bad}} \tilde{f}(S \cup \mathbf{z}) = \max_{\mathbf{z} \in \text{bad}} \xi_{\mathbf{z}} f(S \cup \mathbf{z}) \leq \xi_{\sup}[f(S) + (1 - 3\delta)f_S(\mathbf{x}^\star)]$$

Since $f_S(\mathbf{x}) \in \Omega\left(\frac{f(S)}{k}\right)$, and importantly $k \leq \sqrt{\log n}$ we know that for sufficiently large $n$:

$$\frac{\sqrt{\log n}}{\delta}f_S(\mathbf{x}^\star) \geq f(S).$$

Putting it all together and conditioning on all events we have with probability at least $1 - \frac{4}{\log n}$:

$$
\begin{aligned}
&\tilde{f}(S \cup \hat{\mathbf{x}}) - \tilde{f}(S \cup \mathbf{b}) \\
&\geq \left((1 - \frac{\delta^2}{3\sqrt{\log n}})\xi_{\sup}[f(S) + (1 - 2\delta)f_S(\mathbf{x}^\star)]\right) - \left(\xi_{\sup}[f(S) + (1 - 3\delta)f_S(\mathbf{x}^\star)]\right) \\
&\geq \xi_{\sup}\left(\delta f_S(\mathbf{x}^\star) - \frac{\delta^2}{3\sqrt{\log n}} \times [(1 - 2\delta)f_S(\mathbf{x}^\star) + f(S)]\right) \\
&\geq \xi_{\sup}\left(\delta f_S(\mathbf{x}^\star) - \frac{\delta^2}{3\sqrt{\log n}} \times \left[(1 - 2\delta)f_S(\mathbf{x}^\star) + \frac{\sqrt{\log n}}{\delta}f_S(\mathbf{x}^\star)\right]\right) \\
&> \xi_{\sup} \cdot \frac{\delta}{3}f_S(\mathbf{x}^\star)
\end{aligned}
$$

Since the difference is strictly positive this implies that with probability at least $1 - \frac{4}{\log n}$ a bad bundle will not be selected as $\hat{\mathbf{x}}$, which concludes our proof. $\qquad\square$

## B.1 Optimization for Constant $k$

**Redefining the ball.** For every bundle $\mathbf{x}$ of size $c$ we define the ball $\mathcal{B}(\mathbf{x}) = \{\mathbf{x} \cup z : z \notin \mathbf{x}\}$, and the mean value and noisy mean values are $F(\mathbf{x}) = \mathbb{E}_{\mathbf{z} \in \mathcal{B}(\mathbf{x})}[f(\mathbf{z})]$ and $\tilde{F}(\mathbf{x}) = \mathbb{E}_{\mathbf{z} \in \mathcal{B}(\mathbf{x})}[\tilde{f}(\mathbf{z})]$, respectively. Using the same reasoning as in Lemma 2.1 we get a concentration bound on $\operatorname{argmax}_{\mathbf{b}:|\mathbf{b}|=c} \tilde{F}(\mathbf{b})$.

**Lemma B.1.** *Let $\mathbf{x} \in \operatorname{argmax}_{\mathbf{b}:|\mathbf{b}|=c} \tilde{F}(\mathbf{b})$. Then, for any $\lambda > 0$ w.p. $1 - \exp(-\Omega(\lambda^2\sqrt{n - c}))$:*

$$F(\mathbf{x}) \geq (1 - \lambda)\max_{\mathbf{b}:|\mathbf{b}|=c} F(\mathbf{b}).$$

---
**Algorithm 3** EXP-SM-GREEDY
---
**Input:** budget $k$
  1: $\mathbf{x} \leftarrow \mathrm{argmax}_{\mathbf{b}:|\mathbf{b}|=k} \tilde{F}(\mathbf{b})$
  2: $z \leftarrow$ select random element from $N \setminus \mathbf{x}$
  3: $\hat{\mathbf{x}} \leftarrow$ random set of size $k$ from $\mathbf{x} \cup z$
  4: **return** $\hat{\mathbf{x}}$

---

**Approximation guarantee in expectation.** We first present the algorithm whose approximation guarantee is arbitrarily close to $1 - \frac{1}{k+1}$, in expectation. The algorithm will simply select the set $\hat{\mathbf{x}}$ to be a random subset of $k$ elements from a random set of $\mathcal{B}(\mathbf{x})$ where $\mathbf{x} \in \mathrm{argmax}_{\mathbf{b}:|\mathbf{b}|=k} \tilde{F}(\mathbf{b})$.

**Theorem B.2.** *For any submodular $f : 2^N \to \mathbb{R}$, EXP-SM-GREEDY returns a $(1 - \frac{1}{k+1} - o(1))$ approximation for $\max_{S:|S|\leq k} f(S)$, in expectation, using a generalized exponential tail noisy oracle.*

*Proof.* Similar to Lemma 2.2, by submodularity we know that in expectation $f(\hat{\mathbf{x}}) \geq \frac{k}{k+1} F(\mathbf{x})$. Let $\mathbf{x}^\star = \mathrm{argmax}_{\mathbf{b}:|\mathbf{b}|=k} f(\mathbf{b})$. From monotonicity we know that $f(\mathbf{x}^\star) \leq F(\mathbf{x}^\star)$. Applying Lemma B.1 we get that for the set $F(\mathbf{x}) \geq (1 - o(1))F(\mathbf{x}^\star)$. Hence:

$$\mathbb{E}[f(\hat{\mathbf{x}})] \geq \left(\frac{k}{k+1}\right) F(\mathbf{x}) \geq \left(\frac{k}{k+1} - o(1)\right) F(\mathbf{x}^\star) \geq \left(\frac{k}{k+1} - o(1)\right) f(\mathbf{x}^\star) = \left(\frac{k}{k+1} - o(1)\right) \mathrm{OPT}.$$

$\square$

**Approximation Guarantee with high probability.** To obtain a result that holds w.h.p. we will consider a modest variant of the algorithm above. The algorithm enumerates all possible subsets of size $k-1$, identifies the bundle $\mathbf{x} \in \mathrm{argmax}_{\mathbf{b}:|\mathbf{b}|=k-1} \tilde{F}(\mathbf{b})$ and then returns $\hat{\mathbf{x}} \in \mathrm{argmax}_{\mathbf{z} \in \mathcal{B}(\mathbf{x})} \tilde{f}(\mathbf{z})$.

---
**Algorithm 4** WHP-SM-GREEDY
---
**Input:** budget $k$
  1: $\mathbf{x} \leftarrow \arg\max_{\mathbf{b}:|\mathbf{b}|=k-1} \tilde{F}(\mathbf{b})$
  2: $\hat{\mathbf{x}} \leftarrow \mathrm{argmax}_{\mathbf{z} \in \mathcal{B}(\mathbf{x})} \tilde{f}(\mathbf{z})$
  3: **return** $\hat{\mathbf{x}}$

---

**Theorem B.3.** *For any submodular function $f : 2^N \to \mathbb{R}$ and any fixed $\epsilon > 0$ and constant $k$, there is a $(1 - 1/k - \epsilon)$-approximation algorithm for $\max_{S:|S|\leq k} f(S)$ which only uses a generalized exponential tail noisy oracle, and succeeds with probability at least $1 - 6/\log n$.*

*Proof.* Let $\mathbf{x} \in \mathrm{argmax}_{\mathbf{b}:|\mathbf{b}|=k-1} \tilde{F}(\mathbf{b})$, and let $\mathbf{x}^\star \in \mathrm{argmax}_{\mathbf{b}:|\mathbf{b}|=k-1} f(\mathbf{b})$. Since $\mathbf{x}^\star$ is the optimal solution over $k-1$ elements, from submodularity we know that $f(\mathbf{x}^\star) \geq (1 - 1/k)\mathrm{OPT}$.

What now remains to show is that $\hat{\mathbf{x}} \in \mathrm{argmax}_{z \in N \setminus \mathbf{x}} \tilde{f}(\mathbf{x} \cup z)$ is a $(1 - \epsilon)$ approximation to $F(\mathbf{x})$. To do so, recall the definitions of good and bad bundles from the previous section: let $\delta = \epsilon/3$, and say a bundle $\mathbf{z}$ is *good* if $f(\mathbf{z}) \geq (1 - 2\delta)f(\mathbf{x}^\star)$ and *bad* if $f(\mathbf{z}) \leq (1 - 3\delta)f(\mathbf{x}^\star)$. We show that with high probability the bundle $\hat{\mathbf{x}}$ selected by the algorithm has value at least as high as that of a bad bundle, i.e. $f(\hat{\mathbf{x}}) \geq (1 - 3\delta)f(\mathbf{x}^\star)$ which will complete the proof.

We first show that with probability at least $1 - 6/\log n$ the maximal noise multiplier of a good bundle is at least $\xi_{\inf}$ and of a bad bundle is at most $\xi_{\sup}$, where we use the same definition of $\xi_{\inf}$ and $\xi_{\sup}$ as in Section 3. To do so we will first argue about the number of good bundles in the ball. From Lemma B.1 and the maximality of $\mathbf{x}$ we know that with overwhelming high probability $F(\mathbf{x}) \geq (1 - o(1))F(\mathbf{x}^\star)$. Therefore for $m = n - k$ and fixed $\delta$:

$$F(\mathbf{x}) = \frac{1}{m} \sum_{z \notin \mathbf{x}} f(\mathbf{x} \cup z) \geq (1 - \delta)\frac{1}{m} \sum_{z \notin \mathbf{x}^\star} f(\mathbf{x}^\star \cup z) \geq (1 - \delta)f(\mathbf{x}^\star)$$

Let $\mathcal{B}^+(\mathbf{x})$ be the bundle of all good bundles in $\mathcal{B}(\mathbf{x})$. Due to the maximality of $\mathbf{x}^\star$ and submodularity we know that $f(\mathbf{x} \cup z) \leq 2f(\mathbf{x}^\star)$ for all $z \notin \mathbf{x}$:

$$\sum_{z \notin \mathbf{x}} f(\mathbf{x} \cup z) \leq |\mathcal{B}^+(\mathbf{x})| 2f(\mathbf{x}^\star) + (m - |\mathcal{B}^+(\mathbf{x})|)(1 - 2\delta)f(\mathbf{x}^\star)$$

Putting the these bounds on $F(\mathbf{x})$ together and rearranging we get that:

$$|\mathcal{B}^+(\mathbf{x})| \geq \frac{\delta \cdot m}{1 + 2\epsilon} \geq \frac{\delta m}{3}$$

Since there are at least $\delta m/3$ good bundles we can bound the likelihood of at least one noise multiplier of a good bundle achieving value $\xi_{\inf}$:

$$\Pr\left[\max\{\xi_1, \ldots, \xi_{\delta \cdot m/3}\} \geq \xi_{\inf}\right] \geq 1 - \left(1 - \frac{2\log n}{m}\right)^{\frac{\delta m}{3}} \geq 1 - \frac{2}{n^{\delta/3}} \geq 1 - \frac{1}{\log n}$$

Since there are $m = n - k$ bundles in the ball, the likelihood that all noise multipliers of bad bunldes are bounded from above by $\xi_{\sup}$ is:

$$\Pr\left[\max\{\xi_1, \ldots \xi_m\} \leq \xi_{\sup}\right] \geq \left(1 - \frac{2}{m\log n}\right)^m > \left(1 - \frac{4}{\log n}\right)$$

Thus, by a union bound and conditioning on the event in Lemma B.1 we get that $\xi_{\sup}$ is an upper bound on the value of the noise multiplier of bad bundles and $\xi_{\inf}$ is with lower bound on the value of the noise multiplier of a good stem all with probability at least $1 - 6/\log n$.

We therefore know that with probability at least $1 - 6/\log n$ the maximal noise multiplier of a good bundle is at least $\xi_{\inf}$ and the noise multiplier of a bad bundle is at most $\xi_{\sup}$. From Lemma 3.3 we know that $\xi_{\inf} \geq (1 - o(1))\xi_{\sup}$. Thus:

$$\max_{\mathbf{z} \in \mathcal{B}^+(\mathbf{x})} \tilde{f}(\mathbf{z}) = \max_{\mathbf{z} \in \in \mathcal{B}^+(\mathbf{x})} \xi_{\mathbf{z}} f(\mathbf{z}) \geq \xi_{\inf} \cdot (1 - 2\delta)f(\mathbf{x}^\star) \geq \xi_{\sup} \cdot (1 - 2\delta - o(1))f(\mathbf{x}^\star)$$

Let $\mathbf{b} \in \mathrm{argmax}_{\mathbf{z} \in \text{bad bundles}} \tilde{f}(\mathbf{x} \cup \mathbf{z})$:

$$\tilde{f}(\mathbf{b}) \geq \max_{\mathbf{z} \in \text{bad bundles}} \tilde{f}(\mathbf{z}) = \max_{\mathbf{z} \in \text{bad bundles}} \xi_{\mathbf{z}} f(\mathbf{z}) \leq \xi_{\sup} \cdot (1 - 3\delta)f(\mathbf{z})$$

Putting it all together we have with probability at least $1 - 6/\log n$:

$$\tilde{f}(\hat{\mathbf{x}}) - \tilde{f}(\mathbf{b}) \geq \xi_{\sup} f(\mathbf{x}^\star) \cdot \left((1 - 2\delta - o(1)) - (1 - 3\delta)\right) > \xi_{\sup} f(\mathbf{x}^\star)(\delta - o(1))$$

The difference is strictly positive, and since $\delta = \epsilon/3$ is fixed and this completes our proof. $\qquad\square$

## C  Approximation Algorithm for Matroids

We begin with basic facts and definitions about Matroids and properties of submodular functions.

**Claim C.1.** *Let $f : 2^N \to \mathbb{R}$ be a submodular function and $S_k, O \subseteq N$. Then we have that:*

$$f(O) \leq f(O \cup S_k) \leq f(S_k) + \sum_{x \in O \setminus S_k} f_{S_k}(x) \tag{7}$$

*Proof.* This is a direct consequence of submodularity. $\qquad\square$

**Definition C.2** (rank and span of a matroid)**.** *For a set $S$ and a matroid $\mathcal{M}_j$ in the family $\mathcal{F}$, we define $\mathrm{rank}_j(S)$, called the rank of $S$ in $\mathcal{M}_j$ to be the cardinality of the largest subset of $S$ which is independent in $\mathcal{M}_j$, and define $\mathrm{span}_j(S)$, called the span of $S$ in $\mathcal{M}_j$ by:*

$$\mathrm{span}_j(S) = \{a \in N : \mathrm{rank}_j(S \cup a) = \mathrm{rank}_j(S)\}$$

**Claim C.3.** *Let $S_i, O \in \mathcal{M}_j$ be independent sets where $S_i = \{\hat{\mathbf{x}}_1, \ldots, \hat{\mathbf{x}}_i\}$ is a set of bundles of size $c$ each. Then:*

$$|\mathrm{span}_j(S_i) \cap O| \leq c \cdot i$$

*Proof.* Since $S_i$ is independent in $\mathcal{M}_j$, we know that $\text{rank}_j(\text{span}_j(S_i)) = \text{rank}_j(S_i) = |S_i|$. In particular, we have that $\text{rank}_j(\text{span}_j(S_i)) = c \cdot i$. Since $O$ is an independent set in $\mathcal{M}_j$ we have:

$$\text{rank}_j(\text{span}_j(S_i) \cap O) = |\text{span}_j(S_i) \cap O| \le |\text{span}_j(S_i)| = |S_i| \le c \cdot i$$

where the above inequality is due to the fact that for any independent set $T$ in $\mathcal{M}_j$ we have that $\text{rank}_j(T) = |\text{span}_j(T)| = |T|$. This implies that $|\text{span}_j(S_i) \cap O| \le c \cdot i$. $\qquad\square$

**Claim C.4** (Prop. 2.2 in [NWF78a]). *If for $\forall t \in [k]$ $\sum_{i=0}^{t-1} \sigma_i \le t$ and $p_{i-1} \ge p_i$, with $\sigma_i, p_i \ge 0$ then:*

$$\sum_{i=0}^{k-1} p_i \sigma_i \le \sum_{i=0}^{k-1} p_i.$$

**Lemma.** *4.1 Let $O$ be the optimal solution, $k = |O|$, and for every iteration $i$ of* SM-MATROID-GREEDY *let $S_i$ be the set of elements selected and $\mathbf{x}_i^\star \in \text{argmax}_{|\mathbf{z}|=c} f_{S_{i \cdot 1}}(\mathbf{z})$ be the optimal bundle at stage $i$ of the algorithm. Then:*

$$f(O) \le (P+1) \sum_{i=1}^{\frac{k}{c}} f_{S_i}(\mathbf{x}_i^\star)$$

*Proof.* Since $S_i$ is independent in $\mathcal{M}_j$, we know that $\text{rank}_j(\text{span}_j(S_i)) = \text{rank}_j(S_i) = |S_i|$. In particular, we have that $\text{rank}_j(\text{span}_j(S_i)) = c \cdot i$. Now in each $1 \le j \le P$, since $O$ is an independent set in $M_j$ we have:

$$\text{rank}_j(\text{span}_j(S_i) \cap O) = |\text{span}_j(S_i) \cap O|$$

which by Claim C.3 implies that $|\text{span}_j(S_i) \cap O| \le c \cdot i$.

Define $U_i = \cup_{j=1}^{P} \text{span}_j(S_i)$, to be the set of elements which are not part of the maximization in step $i + 1$ of the procedure, and hence cannot give value at that stage. We have:

$$|U_i \cap O| = |(\cup_{j=1}^{P} \text{span}_j(S_i)) \cap O| \le \sum_{j=1}^{P} |\text{span}_j(S_i) \cap O| \le P \cdot c \cdot i$$

Let $V_i = (U_i \setminus U_{i-1}) \cap O$ be the elements of $O$ which are not part of the maximization in step $i$, but were part of the maximization in step $i - 1$. If $x \in V_i$ then it must be that

$$f_k(x) \le f_{S_i}(x) \le \frac{f_{S_i}(\mathbf{x}_i^\star)}{c}$$

where the first inequality is due to submodularity of $f$ and the second is since $x$ was not chosen in step $i$. Hence, we can upper bound:

$$\sum_{x \in O \setminus S_k} f_{S_k}(x) \le \sum_{i=1}^{\frac{k}{c}} \sum_{x \in V_i} \frac{f_{S_i}(\mathbf{x}_i^\star)}{c} = \sum_{i=1}^{\frac{k}{c}} |V_i| f_{S_i}(\mathbf{x}_i^\star) \le P \sum_{i=1}^{\frac{k}{c}} f_{S_i}(\mathbf{x}_i^\star)$$

where the last inequality uses $\sum_{t=1}^{i} |V_t| = |U_i \cap O| \le Pi$ and the following arithmetic claim due to C.4. Together with (7), we get:

$$f(O) \le (P+1) \sum_{i=1}^{\frac{k}{c}} f_{S_i}(\mathbf{x}_i^\star)$$

as required. $\qquad\square$

# D Information Theoretic Lower Bounds

**Claim D.1.** *There exists a submodular function and noise distribution s.t. no randomized algorithm can obtain an approximation better than $1/2 + O(1/\sqrt{n})$ for $\max_{a \in N} f(a)$ w.h.p. using a noisy oracle.*

*Proof.* We will construct two functions that are identical except that one function attributes a value of 2 for a special element $x^\star$ and 1 for all other elements, whereas the other is assigns a value of 1 for each element. In addition, these functions will be bounded from above by 2 so that the only queries that give any information are those of singletons. More formally, consider the functions $f_1(S) = \min\{|S|, 2\}$ and $f_2(S) = \min\{g(S), 2\}$ where $g : 2^N \to \mathbb{R}$ is defined for some $x^\star \in N$ as:

$$g(S) = \begin{cases} 2, & \text{if } S = x^\star \\ |S|, & \text{otherwise} \end{cases}$$

The noise distribution will return 2 with probability $1/\sqrt{n}$ and 1 otherwise.

We claim that no algorithm can distinguish between the two functions with success probability greater than $1/2 + O(1/\sqrt{n})$. For all sets with two or more elements, both functions return 2, and so no information is gained when querying such sets. Hence, the only information the algorithm has to work with is the number of 1, 2, and 4 values observed on singletons. If it sees the value 4 on such a set, it concludes that the underlying function is $f_2$. This happens with probability $1/\sqrt{n}$.

Conditioned on the event that the value 4 is not realized, the only input that the algorithm has is the number of 1s and 2s it sees. The optimal policy is to choose a threshold, such if a number of 2s observed is or above this threshold, the algorithm returns $f_2$ and otherwise it reruns $f_1$. In this case, the optimal threshold is $\sqrt{n} + 1$.

The probability that $f_2$ has at most $\sqrt{n}$ twos is $1/2 - 1/\sqrt{n}$, and so is the probability that $f_1$ has at least $\sqrt{n} + 1$ twos, and hence the advantage over a random guess is $O(1/\sqrt{n})$ again.

An algorithm which approximates the maximal set on $f_2$ with ratio better than $1/2 + \omega(1/\sqrt{n})$ can be used to distinguish the two functions with advantage $\omega(1/\sqrt{n})$. Having ruled this out, the best approximation one can get is $1/2 + O(1/\sqrt{n})$ as required. $\qquad\square$

We now construct a lower bound for general $k$ of $(2k-1)/2k$, where our upper bound is $(k-1)/k$.

**Claim D.2.** *There exists a submodular function and noise distribution for which w.h.p. no randomized algorithm with a noisy oracle can obtain an approximation better than $(2k-1)/2k + O(1/\sqrt{n})$ for the optimal set of size $k$.*

*Proof.* Consider the function:

$$f_1(S) = \begin{cases} 2|S|, & \text{if } |S| < k \\ 2k-1, & \text{if } |S| = k \\ 2k, & \text{if } |S| > k \end{cases}$$

and the function $f_2$, which is dependent on the identity of some random set of size $k$, denoted $S^\star$:

$$f_2(S; S^\star) = \begin{cases} 2|S|, & \text{if } |S| < k \\ 2k-1, & \text{if } |S| = k, S \neq S^\star \\ 2k, & \text{if } S = S^\star \\ 2k, & \text{if } |S| > k \end{cases}$$

Note that both functions are submodular.

The noise distribution will return $2k/(2k-1)$ with probability $n^{-1/2}$ and 1 otherwise. Again we claim that no algorithm can distinguish between the functions with probability greater than $1/2$. Indeed, since $f_1, f_2$ are identical on sets of size different than $k$, and their value only depends on the set size, querying these sets doesn't help the algorithm (the oracle calls on these sets can be simulated). As for sets of size $k$, the algorithm will see a mix of $2k-1$, $2k$, and at most one value of $4k^2/(k-1)$. If the algorithm sees the value $4k^2/(k-1)$ then it was given access to $f_2$. However, the algorithm will see this value only with probability $1/\sqrt{n}$. Conditioning on not seeing this value, the best policy the algorithm can adopt is to guess $f_2$ if the number of $2k$ values is at least $1 + \frac{\binom{n}{k}}{\sqrt{n}}$, and guess $f_1$ otherwise. The probability of success with this test is $1/2 + O(1/\sqrt{n})$ (regardless of whether the underlying function is $f_1$ or $f-2$). Any algorithm which would approximate the best set of size $k$ to an expected ratio better than $(2k-1)/2k + \omega(1/\sqrt{n})$ could be used to distinguish between the function with an advantage greater than $1/\sqrt{n}$, and this puts a bound of $(2k-1)/2k + O(1/\sqrt{n})$ on the expected approximation ratio. $\qquad\square$

We note that if the algorithm is not allowed to query the oracle on sets of size greater than $k$, Claim D.1 can be extended to show a $\Omega(1/n)$ inapproximability, so choosing a random element is almost the best possible course of action.

# E   From maximizing $f$ to maximizing $\tilde{f}$

Similar to the previous section, let $f$ be a submodular function, let $g$ be a function which is derived from $f$ by sampling for each $x \subset [n]$ a function $h \in_{\mathcal{D}} \mathcal{H}$ and setting $g(x) = h(f(x))$. In this section we assume that the family $\mathcal{H}$ consists of monotone concave functions. We are trying to maximize $g$ under an intersection of matroids $\mathcal{F}$. Suppose that we are allowed unlimited oracle access to $f$, but only $n^c$ oracle invocation of $g$ for some $c > 0$. Let $\mathcal{ALG}(n^c)$ be the following algorithm:

1. Find sets $S_1, S_2, \ldots S_{n^c}$ such that $S_i = \arg\max_{S \in [n], S \in P, S \neq S_1, S_2 \ldots S_{i-1}}$.
2. Output $\arg\max_{S_i} g(S_i)$

**Lemma E.1.** *Algorithm $\mathcal{ALG}(n^c)$ is optimal if we are only allowed $n^c$ oracle invocations of $g$.*

Note that we are not necessarily finding the optimal set, but this is the best one can do in this setting.

To set a more realistic model, let $S* = \operatorname{argmax}_{S \subset [n], \ S \in \mathcal{F}} f(S)$, and suppose that we are given a set $\tilde{S} \in \mathcal{F}, |\tilde{S}| \geq 1.01c \log /(\epsilon \log(1/\epsilon))$ such that $f(\tilde{S}) \geq \alpha f(S*)$ for some $\alpha > 0$.

We are still allowed $n^c$ oracle calls to $g$, but we are not allowed any oracle calls to $f$. Let $\mathcal{ALG}$ be the following algorithm:

1. Find sets $S_1, S_2, \ldots S_{n^c} \subset \tilde{S}$ with no repetitions, such that $S_i$ is chosen at random between all sets with maximal intersection size with $\tilde{S}$.
2. Output $\arg\max_{S_i} g(S_i)$

**Lemma E.2.** *Algorithm $\mathcal{ALG}$ gives an $\alpha(1 - \epsilon)$ approximation to $\mathcal{ALG}(n^c)$.*

*Proof.* Let $\tilde{S}^{-j}$ be all the subsets of $\tilde{S}$ of size $k - j$. We have $|S^{-j}| = \binom{k}{j}$. Let $\alpha = 1.01c \log n / \log(1/\epsilon)$. We claim that in step 1 of $\mathcal{ALG}$ the set $S_{n^c}$ has at least $k - \alpha$ elements. To see this, we let

$$\log(\sum_{j=1}^{\alpha} \binom{k}{j}) \geq \log(\binom{k}{\alpha})$$

$$\geq 0.999kH(\alpha/k) \geq 0.999k(\alpha/k)\log(k/\alpha)$$
$$\geq 0.999\alpha\log(k/\alpha) \geq 0.999\alpha\log(1/\epsilon) \geq c\log n$$

So there are at least $n^c$ different sets created in the first step. As $\alpha \leq |\tilde{S}|/\epsilon$, the expected value of $f(S_i)$ is at least $(1 - \epsilon)f(S)$. The expected value of running $\mathcal{ALG}$ is at least the maximum of $h_1((1 - c/k)f(S)), \ldots h_{n^c}((1 - c/k)f(S))$ where $h_i$ is sampled independently from $\mathcal{H}$ according to $\mathcal{D}$. Since each $h_i$ is convex, in expectation this is at least the maximum of $n^c$ samples of the form $h((1 - \epsilon)f(S))$ where $h$ is sampled independently each time. $\square$

If $\mathcal{H}$ is bounded and independent of $n$, then we get

**Lemma E.3.** *Algorithm $\mathcal{ALG}$ gives an $(1 - 2\epsilon)$ approximation to the optimal value.*

The proof relies on the fact that if one samples $\mathcal{H}$ enough times then one gets a value which is a $1 - \epsilon$ approximation to the optimal value. We note that in this case it is enough to sample $\mathcal{H}$ any super constant number of times (when $n$ is large enough), so we no longer need any requirement on the size of $\tilde{S}$.

# F Extensions

## F.1 Additive approximations

Throughout this paper we assumed the approximation is multiplicative, i.e. we defined the oracle to return $\tilde{f}(S) = \xi_S \cdot f(S)$. An alternative model is one where the approximation is *additive*, i.e. $\tilde{f}(S) = f(S) + \xi_S$, where $\xi_S \sim \mathcal{D}$. We note that the impossibility results for functions that are $\epsilon$-close to submodular also apply to additive approximations.

From a technical perspective, the problem remains non-trivial. Fortunately, all the algorithms described above apply to the additive noise model, modulo the weak concentration bounds which become straightforward:

$$\tilde{F}(S \cup \mathbf{x}) = \frac{1}{|\mathcal{B}_S(\mathbf{x})|} \sum_{\mathbf{z} \in \mathcal{B}_S(x)} \tilde{f}(S \cup \mathbf{z})$$

$$= \frac{1}{|\mathcal{B}_S(\mathbf{x})|} \sum_{\mathbf{z} \in \mathcal{B}_S(\mathbf{x})} (f(S \cup \mathbf{z}) + \xi_{S \cup \mathbf{z}})$$

$$= \frac{1}{|\mathcal{B}_S(\mathbf{x})|} \left( \sum_{\mathbf{z} \in \mathcal{B}_S(\mathbf{x})} f(S \cup \mathbf{z}) + \sum_{\mathbf{z} \in \mathcal{B}_S(\mathbf{x})} \xi_{S \cup \mathbf{z}} \right)$$

Thus, by applying a concentration bound we can show that a bundle $\mathbf{x}$ whose noisy mean value is maximal implies that its non-noisy mean marginal contribution $F_S(\mathbf{x})$ is approximately maximal.

## F.2 Correlated Approximations

No algorithm can optimize a monotone submodular function under a cardinality constraint given access to a approximate oracle whose multipliers are arbitrarily correlated across sets, even when the support of the distribution is arbitrarily small (this is implied from [HS17]). In light of this, one may wish to consider special cases of correlated distributions.

**Guarantees for $d$-correlated distributions.** Our algorithms can be extended to a model in which querying similar sets may return results that are arbitrarily correlated, as long as querying sets which are sufficiently far from each other gives independent answers.

**Definition.** *We say that the samples are $d$-**correlated** if for any two sets $S$ and $T$, such that $|S \setminus T| + |T \setminus S| > d$ we have that the approximation is applied independently to $S$ and to $T$.*

For this model we show that we can obtain an approximation arbitrarily close to $1 - 1/e$ for $O(1)$-correlated noise when $k \geq 2$.

**Modification of algorithms for small $k$ for $O(1)$-correlated noise.** We add $c \gg d/\epsilon$ elements at each phase of the algorithm. We modify the definition of $\tilde{F}$ in the following way. First we take a an arbitrary partition $P_1, \ldots P_{(n-|S|)/d}$ on the elements not in $S$, in which each $P_i$ is of size $d$, and a partition $Q_1 \ldots Q_{(|S|+|A|)/d}$ of the elements in $S \cup A$. We estimate the value of a set $A$ given $S$ using:

$$\tilde{F}(S \cup A) = \frac{d^2}{(|S| + |A|)(|N| - |S| - |A|)} \sum_{Q_i \in A} \sum_{P_j} \tilde{f}(((S \cup A) \setminus Q_i) \cup P_j)$$

and modify the rest of the algorithm accordingly.

Correctness relies on three steps:

1. First, when we are in iteration $i$ of the algorithm (after we already added $(i-1)c$ elements to $S$), all the sets we apply the oracle on are of size $c \cdot i$, and hence they are independent of any set of size $c(i-1)$ or less which were used in previous phases;

2. Second, when we evaluate $\tilde{F}(S \cup A)$ for a specific set $A$, we only use sets which are independent in the comparison. Here we rely on changing $d$ elements in $A$ each time, and replacing them by another set of $d$ elements;

3. Finally, we treat each set $A$ separately, and show that if its marginal contribution is negligible then w.h.p its mean smooth value is not too large, and if its marginal contribution is not negligible, then w.h.p. $\tilde{F}(S \cup A)$ approximates $F(S \cup A)$ well. Taking a union bound over all the bad events we get that the set $A$ chosen has large (non-noisy) smooth mean value.

# G  More related work

The seminal works of [NWF78b, FNW78] show that the greedy algorithm gives a factor of $1 - 1/e$ for maximizing a submodular function under a cardinality constraint and a factor $1/(P+1)$ approximation for intersection of $P$ matroids. Thirty years later Vondrak [Von08] proposed the continuous greedy algorithm, which achieves a $1 - 1/e$ approximation for Matroid constraints. This guarantee is optimal in the value oracle model, even for simple problems like max-cover [Fei98, MSV08, KLMM05, NW78].

Other cases have different approximation constants. One can do better than $1 - 1/f$ with demand queries [FV06]. Non monotone functions have also been studied [FMV11, LMNS09, BFNS12, BFNS14] but we leave approximate versions of such functions to future research. As for techniques for submodular optimization which resemble ours, we note rounding frameworks, optimization of multilinear relaxations and convex relaxations [AS04, CE11, CCPV07, Von08, CJV15, VCZ11].

On the applications front, Submodular utilities have been studied extensively in game theory (e.g. [DNS05, DS06, DLN08, MSV08, BDF$^+$10, DFK11, PP11, DRY11, DV12, PSS08, SS08, BSS10, LSST13]. However, lately it is more and more common to assume that valuations are not exactly submodular [FFI$^+$15]. Similarly, many problems in ML are modeled via submodular functions (e.g. identifying influencers in social networks [KKT03, RLK11] sensor placement [LKG$^+$07, GFK10], learning in data streams [SGK09, GK10, KMVV13, BMKK14], information summarization [LB11a, LB11b], adaptive learning [GK11], vision [JB11b, JB11a, KOJ13], and general inference methods [KG07, JB11a, DK14]). We note that these functions are often inferred from data, and may not be exactly submodular.

We especially mention the notion of *Probably Mostly Approximately Correct* (PMAC) learnability due to Balcan and Harvey [BH11]. Informally, PMAC-learnability guarantees that after observing polynomially-many samples of sets and their function values, one can construct a surrogate function that is with constant probability over the distributions generating the samples, likely to be an approximation of the submodular function generating the data. If we assume that this function is generated in an i.i.d manner we can optimize it.

## G.1  Comparison to with previous work

The algorithms we develop here are very different from the one in [HS17]. At a high level, the idea in [HS17] is to choose an arbitrarily set of elements $H$ of size $\Omega(\frac{\log \log n}{\epsilon^2})$ for a technique called *smoothing*. Smoothing with the set $H$ refers to the process of greedily selecting the element $a \in N \notin H$ that maximizes $\sum_{H' \subseteq H} \tilde{f}(S \cup H' \cup a)$ in every iteration, when $S$ is the solution used in previous iterations. The size of $H$ is determined to be $\Omega(\frac{\log \log n}{\epsilon}^2)$ for concentration bounds to hold. For the approximation guarantees to go through, it is necessary to include the set $H$ in the solution. For this reason, this technique fundamentally requires a cardinality $\Omega(\frac{\log \log n}{\epsilon^2})$ and does not hold for matroids (consider a matroid in which committing to $H$ prevents adding elements with high value).

The algorithm described here adds a constant number of elements at once, and does not commit to selecting any particular set a priori. This lets us prove far stronger results. The new algorithm works for any $k$, and not just for $k \gg \log \log n$. As our lower bounds show, somewhat counter-intuitively, the the smaller cardinality constraint is, the more difficult the problem becomes as there is less room for error. This technique is applicable for intersections of matroids. This relies on not committing to any arbitrary set. We note that randomness would not salvage the algorithm presented in [HS17].