[Reviews · NeurIPS 2018]

Reviewer 1



The authors consider the problem of approximate submodular maximization, i.e. the maximization of functions that can are defined via a submodular function whose set evaluatons are scaled by iid (static) noise. The goal is to maximize the underlying submodular function under cardinality and matroid constraints. The authors develop algorithms for that purpose and proof certain approximation guarantees for them. While the problem itself in its basic version has been studied already, this paper presents improved algorithms and analysis as well as extensions of the considered constraints involving substantial different analysis. I didn't check proofs in details, but on a high-level the author's approach appears to be correct. So novelty and technical contribution certainly make the paper qualify for NIPS. In terms of presentation of the paper, improvements could be made. For instance, I believe that putting parts of appendix E (additive approximation guarantees, correlated approximations, and more connections to related work) would make the paper more compelling than showing partial proofs (I see little value added compared to the high level steps as large parts of the technical contributions are in the appendix anyway). Furthermore, an empirical evaluation would help the reader get an intuition of the importance of parts of the results and where to focus for future algorithmic developments. In particular, the authors argue that they overcome fundamental limitations of previous algorithms, which I understand from their theory, but it would be interesting to see whether this makes a difference in practice. It should be relatively straightforward to implement such an experiment. * f_S(z) seems to be undefined. -- I have read the author response and thank the authors for their comments. I think this is an interesting paper with significant relevance to a group of NIPS attendees.

Reviewer 2



In this paper, the authors study the problem of maximization of approximately submodular functions. They offer good guarantees for both cardinality and general matroid constraints. Here are a few detailed comments, 1. For the three motivations (revealed preference theory etc.) the author mentioned in introduction, the approximated surrogate function is the sum of a submodular function and some error but the problem they study is the product. Can the author give more explanation on the relation between both settings? Or can the author give some example of applications that directly involve the product form? 2. It is very interesting that the guarantee the author showed in the paper, i.e. 1-1/e and 1/(P+1), looks like the classical guarantees of naive greedy on submodular max. Can the algorithms be applied to simple submodular max? 3. What’s is running time of the proposed algorithms? It is better to, at least, have some synthetic dataset experiments and compare with some traditional algorithms (e.g. greedy) for both performance and running time.

Reviewer 3



This is a very theoretically intensive paper on approximating submodular function optimization problems. The authors investigate improved accuracy for solving the submodular function optimization problems through information theoretical approaches. The contributions sound interesting and the problem of study is quite interesting to IJCAI audience. While improving accuracy is one hand of approximating submodular function optimization problems, efficiency is another important aspect of such approximation algorithms. It seems that the authors do not really look into the improved efficiency. In addition, the paper has developed a set of theoretical results; however, it seems that no empirical studies have been conducted to demonstrate the perforation of the approximation algorithms, which may be also important for the approximation. There is no conclusion and future work sections in this paper. Did the authors wrap up this work? Overall an interesting paper may require more polished work.